# Model-Based Policy Adaptation for Closed-Loop End-to-End Autonomous Driving

**Haohong Lin[1], Yunzhi Zhang[2], Wenhao Ding[3], Jiajun Wu[2], Ding Zhao[1]**
[1]CMU,    [2]Stanford,    [3]NVIDIA
haohongl@andrew.cmu.edu

## Abstract

End-to-end (E2E) autonomous driving models have demonstrated strong performance in open-loop evaluations but often suffer from cascading errors and poor generalization in closed-loop settings. To address this gap, we propose Model-based Policy Adaptation (MPA), a general framework that enhances the robustness and safety of pretrained E2E driving agents during deployment. MPA first generates diverse counterfactual trajectories using a geometry-consistent simulation engine, exposing the agent to scenarios beyond the original dataset. Based on this generated data, MPA trains a diffusion-based policy adapter to refine the base policy's predictions and a multi-step Q value model to evaluate long-term outcomes. At inference time, the adapter proposes multiple trajectory candidates, and the Q value model selects the one with the highest expected utility. Experiments on the nuScenes benchmark using a photorealistic closed-loop simulator demonstrate that MPA significantly improves performance across in-domain, out-of-domain, and safety-critical scenarios. We further investigate how the scale of counterfactual data and inference-time guidance strategies affect overall effectiveness.

## 1   Introduction

End-to-End (E2E) autonomous driving models have made impressive strides by integrating perception, prediction, and planning into a unified learning framework [1, 2, 3, 4]. Leveraging large-scale offline driving datasets, E2E models perform well under open-loop evaluation protocols, where the agent passively predicts future behaviors from offline recorded observation sequences. However, these models degrade in closed-loop environments, where minor deviations accumulate over time, leading to compounding errors, distribution shifts, and poor generalization to long-horizon scenarios. This performance gap reveals a core challenge: offline training based on empirical risk minimization does not align with the online objective of maximizing cumulative reward, as is illustrated in Figure 1.

To bridge this gap, recent efforts have turned to evaluating the closed-loop performance of E2E agents. Some open-loop such as NavSim proposes the approximate closed-loop evaluation with a Predictive Driver Model Score in an open-loop evaluation fashion. Other works introduced sensor simulation for the closed-loop evaluation, generating camera views based on diffusion models [5, 6], Neural Radiance Field (NeRF) [7, 8, 9, 10] or 3D Gaussian Splatting (3DGS) [11, 12, 13, 14, 15] that enable photorealistic rendering of novel viewpoints. These tools provide fine-grained control over agent interventions and visual realism, making them promising testbeds for studying failure modes and recovery strategies. Existing works such as VAD [4], VAD-v2 [16], and Hydra-MDP [17] design different scoring mechanisms to select the predicted motions for closed-loop control, yet these works either lack closed-loop evaluation results or are evaluated in a non-photorealistic simulator like CARLA [18]. Most recently, RAD [19] incorporates reinforcement learning and uses 3DGS for online rollouts and evaluation, while the training of PPO agents can be costly, and the value

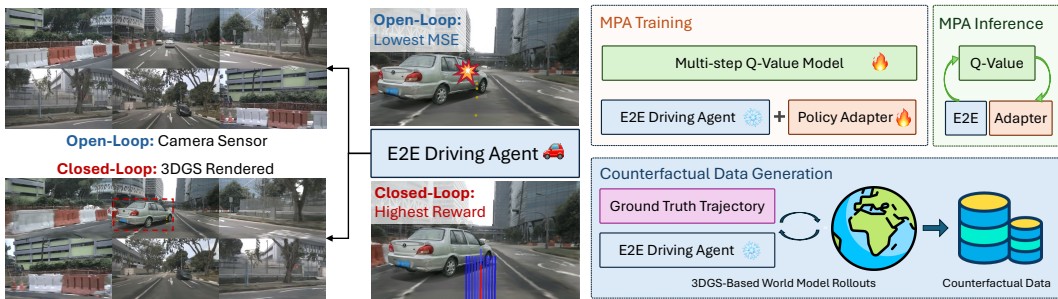

Figure 1: **Left:** Causes of closed-loop performance degradation in End-to-End driving, including observation and objective mismatches. **Right:** We propose counterfactual data generation to address the observation mismatch, and a model-based policy adaptation framework tackling the objective mismatch.

critic is left unused at inference time. Among all the prior attempts, none of the works have curated counterfactual data into consideration during the training phase.

Our goal in this paper is to adapt the pretrained open-loop E2E driving agents from the real domain to safe and generalizable closed-loop agents, with a 3DGS-based driving simulation data engine. We identify that the performance drop between the closed-loop and open-loop evaluations stems from two fundamental sources: (1) Observation mismatch — the shift between training-time sensor inputs and deployment-time closed-loop observations under perturbed behaviors from different data engines; (2) Objective mismatch — the absence of meaningful reward feedback during offline imitation learning, which limits long-term reasoning.

We conducted preliminary experiments to demonstrate that the first mismatch is actually minor in the open-loop evaluation. Then we propose a unified solution called **Model-Based Policy Adaptation (MPA)**, a general framework that directly addresses both mismatches by separating and targeting their root causes. We first use the pretrained policy as an initialization of behavior policy to generate a counterfactual dataset using a high-fidelity 3DGS simulation engine. To mitigate observation mismatch, we design a diffusion-based residual **policy adapter** that conditions on diverse, counterfactual trajectories. This exposes the policy to a broad range of behaviors and visual scenes beyond those seen in the offline dataset. To address objective mismatch, then learn a **Q-value model** from the same counterfactual data that captures long-horizon outcomes and enables value-based assessment beyond rule-based metrics. MPA uses both components at inference time: the policy adapter generates residual action proposals conditioned on the current observation, and the value model performs inference-time scaling to select the action with the highest expected utility.

Our contributions in this work are threefold:

- We analyze the root causes of closed-loop performance degradation in E2E agents and assess the fidelity of 3DGS-based simulation for modeling observation and behavior shifts.
- We develop a systematic counterfactual data curation pipeline using 3DGS rollouts and train the MPA with a diffusion-based policy adapter and reward model to address observation and reward mismatches, respectively.
- We demonstrate that inference-time scaling using the learned reward model significantly improves closed-loop performance on the nuScenes benchmark, particularly in safety-critical and out-of-domain scenarios.

## 2 Preliminary

### 2.1 Problem Formulation of End-to-End Autonomous Driving

We formulate closed-loop end-to-end (E2E) driving as a Partially Observed Markov Decision Process (POMDP) $\mathcal{M} = (\mathcal{S}, \mathcal{A}, P, R, \mathcal{O}, \gamma, T)$, where $\mathcal{S}$ is the latent state space, $\mathcal{A}$ the action space, $P$ the transition dynamics, $R$ the reward function, $\mathcal{O}$ the observation space, $\gamma$ the discount factor, and $T$ the planning horizon. At each timestep, the agent receives an observation $o_t \in \mathcal{O}$ and outputs a trajectory action $a_t \in \mathcal{A}$. The environment evolves according to $P(s_{t+1}|s_t, a_t)$ and emits observations via

$P_{\text{obs}}(o_t|s_t)$. In practice, the state $s_t$ includes the ego vehicle's IMU status and surrounding road entities' past poses and motion intents. These are often only partially observable. The action $a_t$ represents a sequence of future waypoints, which is translated into low-level throttle and steering control sequences via an LQR controller [20], following prior benchmarks [21, 22, 5, 13]. The observation $o_t$ is captured by real sensors or rendered by a simulation engine during closed-loop evaluation. Notably, current open-loop E2E agents are trained using expert trajectories from a behavior policy $\pi_{\text{ref}}$, inducing a state distribution $d^{\text{ref}}(s_t)$ and yielding a supervised model $\hat{\pi}_{\text{ref}}$. In contrast, closed-loop agents aim to maximize cumulative reward over time:

$$
\text{Open-loop}: \hat{\pi}^* = \arg\min_{\pi} \sum_{t=1}^{T} \mathbb{E}_{(s_t,a_t)\sim\pi_{\text{ref}}} \|a_t - \pi(s_t)\|_2^2,
$$

$$
\text{Closed-loop}: \pi^* = \arg\max_{\pi} \sum_{t=1}^{T} \mathbb{E}_{s_t\sim P(s_{t-1},a_{t-1}),\ a_{t-1}\sim\pi(o_{t-1},s_{t-1}),\ o_{t-1}\sim P_{\text{obs}}(s_{t-1})} \left[r(s_t,a_t)\right],
$$

$$
\text{Simulation}: \pi^* = \arg\max_{\pi} \sum_{t=1}^{T} \mathbb{E}_{s_t\sim P(s_{t-1},a_{t-1}),\ a_{t-1}\sim\pi(o_{t-1},s_{t-1}),\ o_{t-1}\sim\widehat{P}_{\text{obs}}(s_{t-1})} \left[r(s_t,a_t)\right].
$$
(1)

This leads to an inherent objective mismatch: open-loop training minimizes imitation error under expert supervision, while closed-loop deployment optimizes long-horizon reward under evolving dynamics and partial observability, as is illustrated in Figure 1. Bridging this gap requires careful alignment of three components in equation (1): the transition model $P(s_{t+1}|s_t, a_t)$, which can be consistently approximated using vehicle dynamics; the observation model $P_{\text{obs}}(o_t|s_t)$, which may deviate from simulated sensors $\widehat{P}_{\text{obs}}$; and the reward function $r(s_t, a_t)$, which must be inferred from partial observations $o_t$ using learned value models.

To address these mismatches, we employ a counterfactual data generation supported by 3DGS-based observation model $\widehat{P}_{\text{obs}}$, then design a policy adapter that transforms the pretrained $\hat{\pi}_{\text{ref}}$ into a reward-aligned policy $\pi^*$ under the guidance of a learned Q-value model, as outlined in Figure 1.

## 2.2 E2E driving Agents in the Closed-loop Simulation

A recent line of approach uses visual generative or reconstruction models for rendering photorealistic driving scenes from state parameterizations, utilizing the recent advancement in image diffusion modeling [5], neural field rendering [10], and 3D Gaussian Splatting [13]. This line of methods essentially learn to estimate $\hat{P}_{\text{obs}}(\cdot|s_t)$ in our POMDP formulation in Section 2.1. However, it is critical to verify that the visual quality of these model outputs remains close to real-world scenes for them to serve as valid proxies for our closed-loop evaluation.

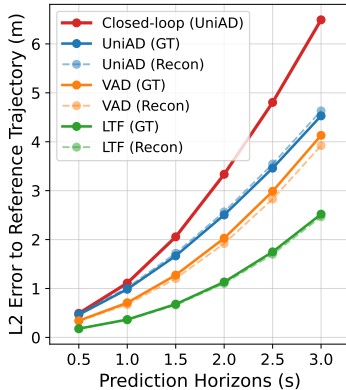

We conducted the following preliminary experiments to study the fidelity of the closed-loop simulator and demonstrate the performance gap between open-loop and closed-loop evaluation. In Figure 2, we systematically study the difference in L2 error under open and closed-loop settings. We use 3DGS [13] to reconstruct the scenes from the nuScenes dataset and compare the performance difference using ground truth data in Figure 2. Among all three E2E policies, we see a very close open-loop performance in motion prediction. This confirms the fidelity of the 3DGS-based simulation in its reconstruction quality.

Figure 2: Comparison of average L2 error in the motion prediction under different prediction horizons.

Meanwhile, we also illustrate the L2 error based on UniAD's closed-loop rollout trajectory. We can see that the prediction error becomes quite significant compared to the open-loop prediction as the prediction horizon grows. A non-ignorable L2 error in the short prediction horizon leads to out-of-distribution issues that result in such compounding errors. Increasing the planning frequency and shortening the planning horizon cannot necessarily close the gap between open-loop and closed-loop performance unless effective feedback guidance is provided to the E2E agents.

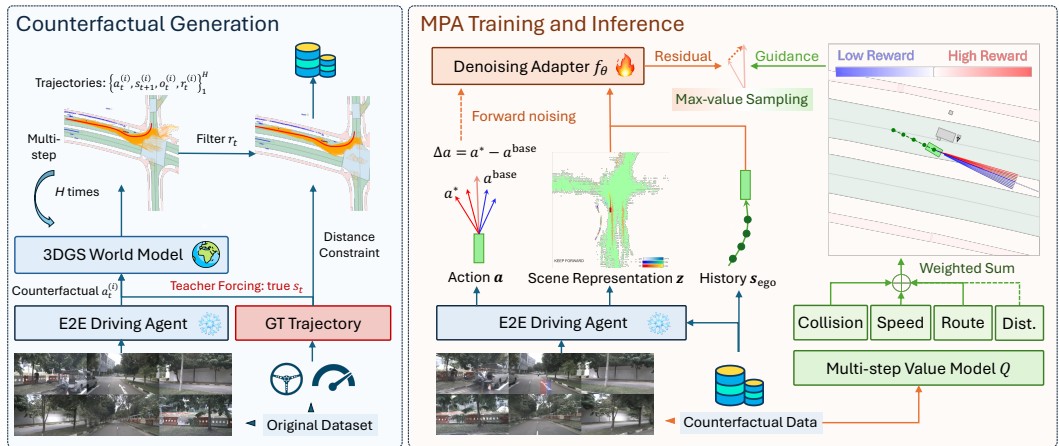

Figure 3: **Overview of Model-Based Policy Adaptation (MPA).** Left: We propose a counterfactual data generation pipeline, where we first generate initial data of $T$-step trajectories rolled out with pretrained E2E policy and 3DGS-based world model, and then filter the generated data with reward and distance constraints to improve data realism, resulting in counterfactual *(action, state, observation, reward)* sequences. Right: We utilize the generated data to train two MPA modules: (i) a diffusion policy adapter predicting residual actions on top of a base E2E agent, and (ii) a value model $Q$ estimating multi-step cumulative rewards under different principles, such as collision and speed.

## 3 Proposed Method: Model-Based Policy Adaptation (MPA)

To bridge the observation and objective mismatches outlined in Section 2.1, we introduce Model-Based Policy Adaptation (MPA)—a unified framework for open-loop to closed-loop adaptation in end-to-end (E2E) autonomous driving. Figure 3 shows an overview. This section is organized from left to right along the pipeline: Section 3.1 describes our model-based counterfactual data synthesis to address distribution mismatch. Section 3.2 details the training of a diffusion-based policy adapter on the curated dataset. Section 3.3 presents the value model used to guide policy adaptation.

### 3.1 World Model-Based Counterfactual Data Generation

We generate counterfactual samples using a geometry-consistent 3D Gaussian Splatting (3DGS) simulator [13], which renders photorealistic observations conditioned on poses of ego vehicle and surrounding agents, modeling the observation distribution $\widehat{P}_{\text{obs}}(\cdot|s_t)$. As shown in Section 2.2, this rendering remains high-fidelity as long as the rollout policy induces a state distribution close to the reference distribution.

To ensure reliable observations while introducing behavioral diversity, we randomly augment the predicted actions from a pretrained E2E policy $\hat{\pi}_{\text{ref}}$ by rotating, warping, and random noising operations, resulting in a noisy policy $\phi_\beta \circ \hat{\pi}_{\text{ref}}$. The resulting distribution of an augmented behavior is shown in Figure 4. Specifically, the augmentation of the trajectory consists of a rotation angle ranging from $[-10, 10]$, and a warping ratio from $[0.1, 1.0]$, as well as random Gaussian noise with standard deviation 0.05. These augmented trajectories will then be categorized into different modes, given their rotation angles and warping ratios for the multi-modal policy adapter. Under a teacher-forcing setup, we initialize the rollout of every driving scene from an original reference state, and conduct counterfactual rollouts. To mitigate the exponentially growing search space as the rollout time horizon expands, we refer to the beam search algorithm [23], maintaining only the most rewarding candidates based on the cumulative reward calculated by the simulator using rule-based heuristics. The specific definition of these rewards can be found in Appendix A. A similar proposal—scoring—selection pipeline is used in the prior literature [24] to mitigate the prediction-planning misalignment of the motion planner.

To prevent rendering artifacts, we discard trajectories that deviate beyond a distance threshold or fall below a minimum reward. The rollout horizon $T$ determines how far into the future these

counterfactual behaviors extend; as we show in our experiments, longer horizons expose richer supervision signals for downstream learning, b ut increase the risk of divergence from reference data.

The full generation procedure is summarized in Algorithm 1. With the generated counterfactual dataset at hand, we then conduct policy learning and value learning in the following sub-sections.

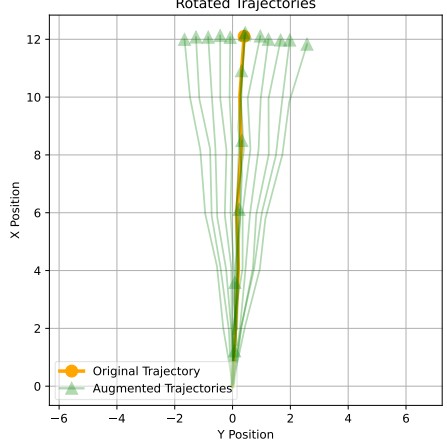

Figure 4: Ego's augmented trajectories after rotating, warping, and random noising operations.

**Algorithm 1:** 3DGS-Based Counterfactual Generation

**Input:** Ref. dataset $\mathcal{D}^{\text{ref}}$, policy $\hat{\pi}_{\text{ref}}$, horizon $T$, thresholds $\delta$, $r_c$
**Output:** Counterfactual dataset $\mathcal{D}^{\text{cf}}$
**foreach** *ref. traj* $(s_0^{ref}, s_1^{ref}, \dots)$ *in* $\mathcal{D}^{ref}$ **do**
    $s_0 \leftarrow s_0^{\text{ref}}$
    **for** $t = 0$ **to** $T-1$ **do**
        $a_t \sim \phi_\beta \circ \hat{\pi}_{\text{ref}}(s_{0:t})$
        **if** $dist(s_t, s_t^{ref}) > \delta$ ***or*** $r(s_t, a_t) < r_c$
        **then**
           | **break**
        $s_{t+1}, r_t \leftarrow \text{Sim.step}(s_t, a_t)$
        $o_t \leftarrow \text{3DGS.Render}(s_t)$
        Append $(s_t, a_t, o_t, r_t)$ to $\mathcal{D}^{\text{cf}}$

## 3.2 Diffusion-Based Policy Adaptation

As we can see in the Figure 4, the generated counterfactual actions fall into a multi-modal distribution. To capture these counterfactual actions, we propose a diffusion-based policy adapter that refines the output of a frozen end-to-end (E2E) driving model by predicting residual trajectories $\Delta a = a^* - a^{\text{base}}$, where $a^{\text{base}} \in \mathbb{R}^{H \times 2}$ is the trajectory from a pretrained policy (e.g., UniAD) and $a^*$ is the most rewarding trajectory sample from the counterfactual rollout dataset.

**Training.** To model the distribution over residuals, we apply a latent diffusion process. The forward step adds Gaussian noise over $K$ steps:

$$q(\Delta a^{(k)} \mid \Delta a^{(0)}) = \mathcal{N}\left(\Delta a^{(k)}; \sqrt{\bar{\alpha}_k}\Delta a^{(0)}, (1 - \bar{\alpha}_k)\mathbf{I}\right),$$

where $\Delta a^{(0)} = \Delta a$ and $\bar{\alpha}_k$ is the cumulative noise schedule. The denoising network $f_\theta$ is a 1D U-Net that outputs multi-mode $\Delta a^{(0)}$ from $\Delta a^{(k)}$, conditioned on the scene encoding $z = \phi_{\text{enc}}(o, \boldsymbol{s}_{\text{ego}})$, ego history $\boldsymbol{s}_{\text{ego}}$, and base predicted trajectory $a^{\text{base}}$. The output of $f_\theta$ contains $N$ modes, we denote the $i$-th mode as $f_\theta(...)[i]$. It is trained with the loss:

$$\mathcal{L}_{\text{diff}} = \mathbb{E}_{\Delta a^{(0)}, k, \epsilon} \min_i \left\| f_\theta(\Delta a^{(k)}, k, z, \boldsymbol{s}_{\text{ego}}, a^{\text{base}})[i] - \Delta a^{(0)} \right\|_2^2,$$

where $\Delta a^{(k)} = \sqrt{\bar{\alpha}_k}\Delta a^{(0)} + \sqrt{1 - \bar{\alpha}_k}\epsilon$, with $\epsilon \sim \mathcal{N}(0, \mathbf{I})$.

**Inference.** For inference, we use DDIM [25] to sample $\Delta a^{(0)}$ and recover the adapted trajectory:

$$a^{\text{adapt}} = a^{\text{base}} + \Delta a^{(0)}.$$

This design adapts the knowledge from the pretrained base policy by conditioning on their encoded scene context and base actions, enabling generalization beyond the counterfactual data domain.

## 3.3 Principled Q-Value Guided Inference-time Sampling

**Training.** While step-wise rewards can be computed given full access to $s_t$, estimating long-horizon returns is challenging under partial observability. Using the generated counterfactual dataset, we train a multi-step action value model

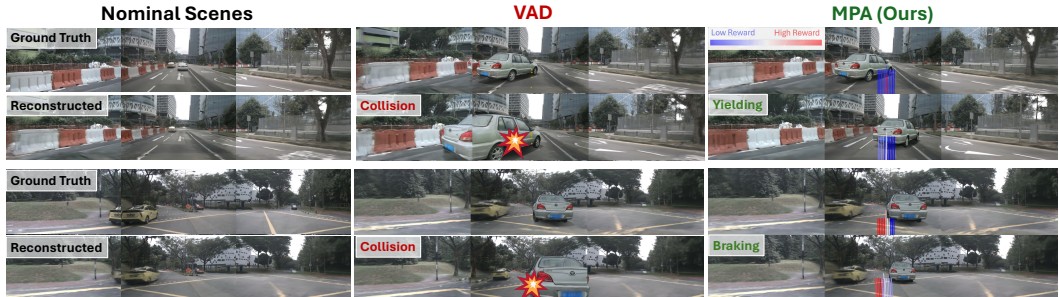

Figure 5: **Qualitative Results** in the in-domain and safety-critical scene. The silver car cuts in from the right side, forcing the ego vehicle to yield. Compared to the pretrain VAD, MPA-adapted policy can successfully brake and yield to the cut-in vehicles under the guidance of the Q-value model.

$$Q(o_t, s_t, a_t; T) = \sum_{t=1}^{T} \gamma^t r(s, a_t)$$

based on four interpretable principles: route following $Q_{\text{route}}$, lane distance $Q_{\text{dist}}$, collision avoidance $Q_{\text{collision}}$, and speed compliance $Q_{\text{speed}}$. Each Q-function is trained independently to predict cumulative returns using $(o_t, s_{\text{ego}}, a_t)$. $Q = \sum_{i \in \{\text{collision, dist, ...}\}} w_i \times Q_i$. We follow the prior literature [26] to design our reward principles and their corresponding weights, with more details in Appendix A. We provide thorough ablations of the $Q$-function components in Section 4.3.

**Inference.** At inference, residual actions $\Delta a$ are sampled following the policy adapter inference procedure from Section 3.2, and the best proposal is selected via

$$\Delta \hat{a}^* = \arg\max_{\Delta a \in a^{\text{adapt}}} Q(o_t, s_{\text{ego}}, a^{\text{base}} + \Delta a; T), \quad \hat{a}^* = a^{\text{base}} + \Delta \hat{a}^*.$$

Compared to classifier-based reward guidance [27, 28], our Q-value guidance offers feedback from longer planning horizons and avoids the gradient instability of the reward model.

## 4 Experiment Results

In our experiments, we aim to answer the following research questions. **RQ1**. Compared to the baselines, can MPA bring benefits to the E2E driving agents in a closed-loop evaluation in a generalizable way? **RQ2**. How does MPA benefit the safety-critical performance in the closed-loop evaluation? **RQ3**. How do different adapters and value guidance modules contribute to the performance of MPA? **RQ4**. How does MPA scale with the number of counterfactual planning steps in the data generation phase?

### 4.1 Experiment Settings

**Dataset and Simulation Engine.** We utilize the nuScenes dataset [29] that consists of 5.5 hours of driving data in Boston and Singapore. Every scene has a reference trajectory of 20 seconds. We use HUGSIM [13] as the simulation engine and evaluation benchmark. We train on a split of 290 scenes in the nuScenes train-val split, and evaluate on three settings. (1) **In-domain evaluation**: the model will be tested on a sub-split of 70 scenes, the surrounding dynamic entities (vehicles, pedestrians) will be replayed by a fixed ratio of their reference trajectory in the offline dataset. (2) **Unseen nominal scene evaluation**: the model will be tested on a sub-split of 70 scenes that are unseen yet during training, the surrounding dynamic entities (vehicles, pedestrians) are nominal and will be replayed by a fixed ratio of their reference trajectory in the offline dataset. (3) **Safety-critical evaluation**: the model will be tested on 10 scenes, where there exists one (or few) non-native agents to challenge the ego agents in an adversarial way. The simulation frequency is 4 Hz. In all the scenes, the termination occurs under one of the following five conditions: (i) full route completion, (ii) off-road events, (iii) collision events, (iv) too far from the reference trajectory, (v) maximum rollout time limits (50 seconds, $2.5\times$ of the reference trajectory) reached.

Table 1: **In-Domain Closed-Loop Evaluation Results.** All the evaluation metrics are higher the better. **Bold** means the best, and underlined is the best runner-up for each metrics.

| Model | Ego Status | Camera | Curation | RC | NC | DAC | TTC | COM | HDScore |
|---|---|---|---|---|---|---|---|---|---|
| UniAD | ✓ | ✓ | ✗ | 39.4 | 56.9 | 75.1 | 52.1 | **98.7** | 19.4 |
| VAD | ✓ | ✓ | ✗ | 50.1 | 68.4 | 87.2 | 66.1 | 90.2 | 31.9 |
| LTF | ✓ | ✓ | ✗ | 65.2 | 71.3 | 92.1 | 67.6 | 98.4 | 46.7 |
| AD-MLP | ✓ | ✗ | ✓ | 13.4 | **80.2** | 86.2 | **79.4** | 90.1 | 6.5 |
| BC-Safe | ✓ | ✓ | ✓ | 57.0 | 59.8 | 87.9 | 55.2 | 89.4 | 33.6 |
| Diffusion | ✓ | ✓ | ✓ | 71.8 | 67.4 | 88.1 | 64.5 | 91.5 | 45.1 |
| MPA (UniAD) | ✓ | ✓ | ✓ | 93.6 | 76.4 | 92.8 | 72.8 | 91.8 | 66.4 |
| MPA (VAD) | ✓ | ✓ | ✓ | **94.9** | 75.4 | **93.6** | 72.5 | 92.8 | **67.0** |
| MPA (LTF) | ✓ | ✓ | ✓ | 93.1 | 70.8 | 90.9 | 67.9 | 94.9 | 60.0 |

**Baselines.** We compare the MPA with diverse baselines in E2E driving algorithms that fall in the two following categories. (1) Pretrained base policy with open-loop training manner: we compare with the performance of **UniAD** [3], **VAD** [4] and **LTF** [2] on the HUGSIM dataset. We further build on our MPA with these policies. (2) E2E agents trained with curated counterfactual dataset: We further train several baseline policies with the curated dataset. **AD-MLP** [30] utilizes the ego's velocity, acceleration, past trajectories, and high-level command as the input, which is recognized as a naive baseline for the closed-loop Driving tasks. **BC-Safe** [31] uses the safe segments in the counterfactual datasets to train and End-to-End policies. **Diffusion** adopts the implementation of [32] in the scene encoding and utilizes a DDIM-based sampler [25] instead of truncated denoising during the inference time. To ensure a fair comparison between the MPA and the second category of the baselines, all the approach uses pretrained ResNet [33] as the perception backbone to encode the RGB inputs from 6 perspective cameras. The ego status includes velocity, acceleration, and navigation landmarks from the history frames.

**Metrics.** We follow the evaluation protocol in HUGSIM [13], which is inspired by the NAVSIM-based metrics [22]. The metrics include Route Completion (RC), Non-Collision (NC), Driveable Area Compliance (DAC), Time-To-Collision (TTC), Comfort (COM), HUGSIM Driving Score (HDScore). Specifically, HDScore is computed with the above metrics along with Route Completion (RC), instead of the Ego Progress (EP) in PDMS [22]. HDScore is the weighted sum as follows:

$$\text{HDScore} = \text{RC} \times \frac{1}{T} \sum_{t=0}^{T} \left\{ \prod_{m \in \{\text{NC, DAC}\}} \text{score}_m \times \frac{\sum_{m \in \{\text{TTC, COM}\}} \text{weight}_m \times \text{score}_m}{\sum_{m \in \{\text{TTC,COM}\}} \text{weight}_m} \right\}_t.$$

We list all the metrics with ($\times 100$) in the tables. All the metrics fall in $[0.0, 100]$. We attach more details of these metrics and other information settings in the Appendix B.1.

## 4.2 Main Results and Analysis (RQ1, RQ2)

To answer RQ1, we first evaluate the closed-loop performance for in- and out-of-domain scenes, as shown in Figure 5. All the reported MPA approaches are evaluated with 20 action samples at inference time. In-domain scenes refer to the scenes that are used to generate counterfactual training data in Singapore. We evaluate the quantitative closed-loop results in these training scenes in Table 1. MPA-based E2E driving agents achieved better results compared to their pretrained counterparts, as well as three baseline methods trained on the counterfactual curated dataset, especially in the most important metrics, RC and HDScore. The baseline AD-MLP moves very conservatively, so the NC and TTC are low, yet the RC is also quite low as it barely completes the assigned routes for the challenging E2E scenes. Besides, the NC score in HUGSIM is a bit underestimated compared to NAVSIM, as HUGSIM erodes the vehicle boxes compared to the actual size by the point clouds. This leads to a few false 'collision' signals during the evaluation. Yet, most of them will not cause a collision that terminates the entire episode in the closed-loop simulation. This is why we observe some high RC with mediocre NC metrics. This still means the ego agents are capable of navigating around different collisions and off-road maneuvers to reach the goal in a reasonable way, and will result in a good HDScore.

We further evaluate the closed-loop performance under the unseen scenes. We select 70 scenes in Boston that are not accessible in the curated counterfactual dataset. The qualitative results are shown

Table 2: **Out-of-Domain Closed-Loop Evaluation Results** in unseen nominal and safety-critical scenes. All the evaluation metrics are higher the better. **Bold** means the best, and underlined is the best runner-up for each metrics.

| Model | Unseen Nominal Scenes | | | | | | Safety-Critical Scenes | | | | | |
|---|---|---|---|---|---|---|---|---|---|---|---|---|
| | RC | NC | DAC | TTC | COM | HDScore | RC | NC | DAC | TTC | COM | HDScore |
| UniAD | 39.3 | 56.6 | 74.0 | 52.6 | **98.2** | 22.2 | 11.4 | 76.2 | 82.1 | 57.8 | 95.9 | 4.5 |
| VAD | 45.4 | 64.8 | 86.2 | 62.0 | 95.9 | 29.3 | 25.4 | 77.0 | 88.3 | 73.2 | 88.4 | 16.0 |
| LTF | 63.3 | 64.8 | 86.5 | 62.8 | **98.2** | 41.9 | 35.1 | 80.9 | 96.8 | 78.1 | **100.0** | 24.2 |
| AD-MLP | 7.6 | **71.6** | 82.2 | **69.8** | 92.3 | 3.3 | 4.9 | **93.5** | 96.2 | **93.4** | 85.9 | 4.3 |
| BC-Safe | 59.2 | 59.8 | 81.2 | 56.3 | 95.9 | 34.6 | 20.2 | 80.1 | 91.7 | 67.3 | 86.7 | 13.5 |
| Diffusion | 57.9 | 62.1 | 83.5 | 58.3 | 96.2 | 35.1 | 20.9 | 84.3 | 92.3 | 72.4 | 86.3 | 13.1 |
| MPA (UniAD) | **93.7** | 69.5 | 92.9 | 66.6 | 97.6 | 60.9 | 95.1 | 76.8 | 98.9 | 74.2 | 97.7 | 70.4 |
| MPA (VAD) | 90.9 | 71.0 | **94.4** | 68.8 | 97.7 | **61.2** | **96.6** | 79.8 | **99.0** | 77.3 | 97.7 | **74.7** |
| MPA (LTF) | 91.8 | 68.3 | 91.0 | 66.5 | 96.9 | 57.0 | 87.3 | 72.0 | 94.0 | 66.9 | 97.8 | 56.3 |

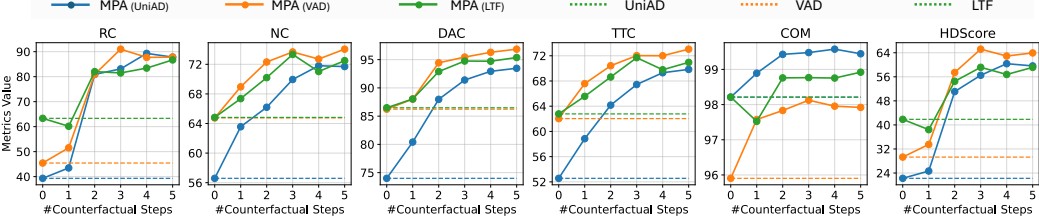

Figure 6: **Impact of Rollout Steps** ($T$ from Algorithm 1) during the counterfactual data generation. We fix the sample size to six during the closed-loop evaluation for all the MPA variants. MPA trained with more counterfactual steps results in better test-time performance, as the Q value model takes more future steps as the supervision signals.

in Table 7 (left). Compared to the in-domain results in Table 1, we can see a significantly degraded performance in AD-MLP and Diffusion, while the pretrained E2E policies still perform similarly as they do in the in-domain scenes. We observe that the MPA agents built upon the pretrained E2E policies are still optimal and quite robust under the unseen scenes. All three variants have comparable HDScore with their in-domain evaluation. This demonstrates the generalizability of the proposed adapter and value model under unseen scene contexts.

### 4.3 Ablation Studies (RQ3, RQ4)

We conduct ablation studies to analyze the contribution of the three main modules of MPA: (i) counterfactual dataset generation, (ii) policy adapter, and (iii) Q-value guidance.

**Counterfactual Dataset.** We further analyze the impact of the curated counterfactual dataset by ablating the step size during the rollout of counterfactual data. Then we train the MPA over the curated dataset and evaluate its performance in the unseen scenes in Boston, similar to the setting in Table 7 (left). We illustrate the trends of evaluation metrics with respect to the step sizes in the counterfactual dataset in Figure 6. MPA benefits from longer counterfactual steps, as there would be more informative supervision for the value function training in the future steps.

**Policy Adapter.** In Table 3, we evaluate a few variants of MPA (UniAD). Due to the space limits, the results of ablation studies for MPA (VAD) and MPA (LTF) will be postponed to the Appendix B. We evaluate the unseen scene's results across different variants. The comparison between ID-5 and ID-6 (ours) demonstrates the effectiveness of the adapter in a better route completion. Under the safety-critical scenes, the adapter brings ∼20% increase to the out route completion, leading to a significantly higher HDScore compared to the policies without the adapter. We also investigate the impact of mode capacity of policy adapter to the driving performances in safety-critical testing splits in Figure 7. The results show that a smaller capacity will influence the DAC and RC metrics, leading to a lower HDScore.

**Principles of Q-Value Guidance.** In Table 3, we remove different principles in the state-action value function $Q$. Compared to the ID-5 variants, ID-1 removes the route information used in all the

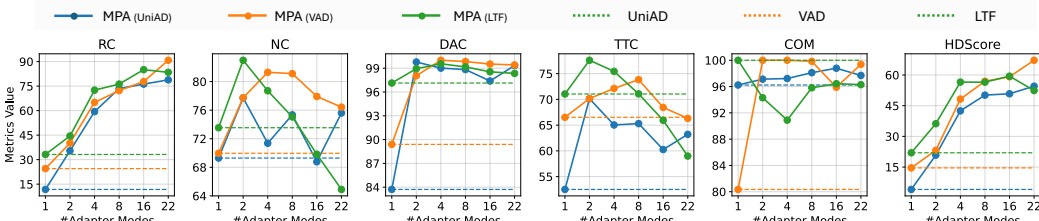

Figure 7: **Impact of Adapter Mode Scales on the Performance under Safety-Critical Scenarios.** The results show that larger sizes of modalities consistently bring benefits to the driving performance in RC, DAC, and HDScore, and the benefit is especially large under the smaller mode sizes.

Table 3: **Ablation Study** on MPA's variants on the UniAD base policy. **Top**: Unseen scenes that are nominal but not appearing in the training dataset. **Bottom**: Safety-critical scenes with adversarial surrounding agents. **Bold** means the best, and underlined is the best runner-up for each metrics.

| ID | $Q_{\text{route}}$ | $Q_{\text{dist}}$ | $Q_{\text{collision}}$ | $Q_{\text{speed}}$ | Adapter | RC | NC | DAC | TTC | COM | HDScore |
|----|----|----|----|----|----|----|----|----|----|----|----|
| 1 | | ✓ | ✓ | ✓ | | 6.9 | 81.2 | 95.1 | 81.0 | 100 | 5.1 |
| 2 | ✓ | | ✓ | ✓ | | 83.9 | 57.0 | 81.0 | 53.6 | 99.4 | 43.2 |
| 3 | ✓ | ✓ | | ✓ | | 89.2 | 70.8 | 95.6 | 68.6 | 99.4 | 60.8 |
| 4 | ✓ | ✓ | ✓ | | | 90.4 | 68.9 | 91.8 | 65.4 | 99.4 | 56.6 |
| 5 | ✓ | ✓ | ✓ | ✓ | | 91.1 | 71.5 | 94.1 | 69.4 | 99.4 | 60.9 |
| 6 | ✓ | ✓ | ✓ | ✓ | ✓ | 93.7 | 69.5 | 92.9 | 66.6 | 97.6 | 60.9 |

| ID | $Q_{\text{route}}$ | $Q_{\text{dist}}$ | $Q_{\text{collision}}$ | $Q_{\text{speed}}$ | Adapter | RC | NC | DAC | TTC | COM | HDScore |
|----|----|----|----|----|----|----|----|----|----|----|----|
| 1 | | ✓ | ✓ | ✓ | | 4.6 | **86.0** | 98.3 | **79.3** | 90.1 | 3.6 |
| 2 | ✓ | | ✓ | ✓ | | 65.1 | 65.6 | 85.7 | 53.8 | 86.5 | 39.5 |
| 3 | ✓ | ✓ | | ✓ | | 57.7 | 82.4 | **99.0** | 69.6 | 84.6 | 39.2 |
| 4 | ✓ | ✓ | ✓ | | | 79.3 | 82.9 | 98.5 | 68.0 | 93.9 | 50.1 |
| 5 | ✓ | ✓ | ✓ | ✓ | | 75.6 | 81.2 | 98.8 | 78.6 | **99.7** | 55.3 |
| 6 | ✓ | ✓ | ✓ | ✓ | ✓ | **95.1** | 76.8 | 98.9 | 74.2 | 97.7 | **70.4** |

baselines, which leads to drastically degraded performance. ID-2 removes the distance function to the reference route, significantly degrading driveable area compliance and non-at-fault collision metrics. ID-3 removes the collision values, and ID-4 removes the speeding value function. Both of them have an impact on HDScore, especially in safety-critical situations. The reason that NC still seemed to be high for ID-3 in safety-critical scenes is that the available frame length before collision is short, which makes the denominator in NC smaller compared to the other group. However, when we look at the RC metrics, the drop when removing $Q_{\text{collision}}$ is significant, as the agents will encounter a collision and end to episode earlier than the nominal cases.

## 5 Related Works

**End-to-End Autonomous Driving.** End-to-End (E2E) autonomous driving has achieved significant progress by jointly training the detection, tracking, prediction, and planning modules to avoid information loss throughout the cascading system. ST-P3 [1] and UniAD [3] propose unified E2E frameworks that achieve state-of-the-art open-loop performance on the nuScenes dataset [29]. VAD [4] encodes the driving scene with a vectorized representation and incorporates query-based planning modules, and VAD-v2 [4] further designs a probabilistic planning approach and improves the closed-loop performance over the CARLA [18] benchmark. LAW [34] enhances VAD's performance with auxiliary supervision signal from latent world model. Hydra-MDP [17] follows VAD-v2's query-based framework and conducts multi-target hydra distillation with a set of scoring rules. SparseDrive [35] uses a sparse-centric pipeline to get rid of BEV features in the E2E driving pipeline, and DiffusionDrive [32] designs a truncated denoising scheme for the diffusion model to generate multi-mode motion trajectories. With the prosperity of foundation models, a series of works [36, 37, 38, 39, 40] incorporate Large Language Models (LLMs) and Vision-Language Models (VLMs) into the E2E planning pipeline. Despite the benefits of commonsense reasoning with foundation models, most of the existing models still focus on the open-loop evaluation or

the approximate closed-loop metrics from the open-loop evaluation [22], which lack counterfactual reasoning for the safety-critical scenarios. A recent work RAD [19] pays attention to the photorealistic closed-loop evaluation and utilizes imitation learning and online reinforcement learning to fine-tune the E2E driving agents. Yet, the value functions trained for the Proximal Policy Optimization (PPO) agents are not effectively integrated during the inference. MPA aims to use the value model as an effective inference-time guidance to make the E2E agents more robust in closed-loop evaluation.

**Counterfactual Data Generation.** Counterfactual data generation has been explored within the context of offline reinforcement learning. Wang et al. [41] utilize a learned model to autonomously generate additional offline data, thereby enhancing the training of sequence models. OASIS [42] introduces a method to produce counterfactual data by modulating guidance signals during diffusion model inference. In high-stakes decision-making domains such as autonomous driving, generating counterfactual data is essential due to the limited presence of safety-critical scenarios in existing datasets. Previous research has addressed the trade-off between *realism* and *controllability* in safety-critical scenario generation by integrating various constraints. These include inference-time sampling techniques [43], retrieval-augmented generation [44], low-rank fine-tuning approaches [45], and language-conditioned generation methods [46]. However, these efforts primarily focus on behavioral scenario generation without incorporating visual information. With advancements in Neural Radiance Fields (NeRF) and 3D Gaussian Splatting (3DGS), recent studies such as DriveArena [5] and MagicDrive [47] have begun developing E2E simulators for closed-loop evaluation. Similarly, RAD [19] employs a 3DGS-based simulator for RL fine-tuning. Notably, to date, no existing work has focused on generating E2E counterfactual data within E2E simulators.

**Reward Model for Inference-Time Scaling.** Recent LLM research has shown the power of the reward model in LLM's inference-time scaling [48, 49]. In sequential decision-making problems, the reward model was explicitly used for inference-time supervision signals, such as the guidance for the diffusion-based policy models [50, 51, 52]. For the closed-loop autonomous driving and decision-making tasks, several prior works incorporate reward models as the classifier-based guidance to steer the diffusion model's sampling process [53, 54, 28]. Other format of the guidance signals include signal temporal logic (STL) guidance, language-based guidance [55], adversarial guidance [56, 57, 58], and game-theoretic guidance [59]. In the autonomous driving domain, prior works like DiffusionDrive [32] utilize truncated denoising for the diffusion models without additional classifier guidance. DiffAD utilizes action conditional guidance for E2E driving [60]. Diffusion-ES [27], Gen-Drive [61], and Diffusion-Planner [28] utilize customized reward models as test-time guidance for the non-E2E driving tasks. To the best of our knowledge, MPA is the first work incorporating the driving reward model for the inference-time scaling of E2E driving agents.

## 6 Conclusion

In this work, we introduce MPA, a general framework for improving the closed-loop trustworthiness of E2E autonomous driving agents. MPA begins by generating high-quality counterfactual trajectories through geometry-consistent rollouts in a 3DGS-based simulation environment. This results in a better data coverage while preserving visual fidelity. With the counterfactual dataset, MPA further trains a diffusion-based policy adapter to refine base policy predictions and leverages a multi-principle value model to guide inference-time decision making. These components allow pretrained agents to generate and evaluate multiple trajectory proposals, selecting actions that optimize long-term driving outcomes. Experimental results on nuScenes data and HUGSIM benchmark demonstrate MPA's effectiveness in boosting safety and generalizability.

**Limitations.** Despite these promising results, our approach assumes reliable rendering from 3DGS under constrained trajectory deviations and currently decouples value modeling from policy optimization. Future work includes extending our current results to diverse driving datasets, exploring the online RL training over the 3DGS simulator, and deploying MPA to the multi-modal foundation models to enhance reasoning capability for more severe distribution shifts in autonomous driving.

## Acknowledgments and Disclosure of Funding

This work is in part supported by ONR MURI N00014-24-1-2748, NSF CNS-2047454, and CMU Safety 21 of the University Transportation Program. YZ is in part supported by the Stanford Interdisciplinary Graduate Fellowship.

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

# A   Auxiliary Details of MPA Implementation

## A.1   Data Collection and Preprocessing

The environments we use to rollout counterfactual data and policy evaluation stay the same with HUGSIM benchmark [13].

**State, Action, and Observation Space of Driving Simulator.**   The driving simulator we use [13] consists of different entities, which will be input to the MPA.

- **Observation (RGB):** The rendered 6 camera views have a resolution of $450 \times 800$ each. This results in a $(6, 3, 450, 800)$ tensor.
- **State and Action (Trajectory):** The pose of every agent can be represented as a tuple $(x, y, yaw)$. We also include the velocity, acceleration, and route information (represented as longitude progress and lateral distance) in the past frames. For the state in HUGSIM, the ego's history trajectory will be a $(T_{\text{hist}}, 8)$ array. For the action output from the agents, it will be a $(T_{\text{fut}}, 2)$ trajectory that consists of the 2D position at each frame. This trajectory will be solved by an LQR controller [20] to generate the corresponding throttle and steering sequence.
- **Intermediate Representation:** all three methods will have an intermediate representation as BEV, where UniAD and LTF have a rasterized map, and VAD has a vectorized map.

**Heuristics-Based Reward Labeling.**   The step-wise reward heuristics include the longitude reward (route completion), lateral distance reward (distance), collision penalty, and far from trajectory penalty. The Q-value is the truncated cumulative sum of the $T$-step reward, where $T$ is the planning horizon as was set up to 5 steps (1.25 s) in our experiments.

$$Q_t = \sum_{k=0}^{T} \gamma^k r_{t+k}$$

Based on these principles, we use the following reward heuristics to label the state-action transition pairs in the simulation. The first term is the longitude reward, denoted as route completion $r_{\text{route}}$. At timestep $t$,

$$r_{\text{route}} = d_t - d_{t-1},$$

where $d_t$ be the projected longitudinal distance along the route from the starting point to the closest point on route to the current ego position determined $\text{pos}_{\text{ego}}$ by $s_t$.

The second term is the lateral distance penalty between the ego position and the route, denoted as distance reward $r_{\text{dist}}$:

$$r_{\text{dist}} = -\|\text{pos}_{\text{ego}} - \text{pos}_{\text{route point}}\|_2.$$

Following the simulation setup, the off-road and collision penalty will be -100 respectively, once a collision between the ego vehicle and other traffic agents or between the ego vehicle and background static objects occurs. This penalty term can be described as:

$$r_{\text{collision}} = \begin{cases} -1, & \min |\text{pos}_{\text{ego}} - \text{pos}_{\text{others}}| < \delta_{\text{collision}}, \\ -1, & \text{pos}_{\text{ego}} \notin \text{Driveable Area}, \\ 0, & \text{otherwise.} \end{cases}$$

To improve the smoothness of driving, we use the speed penalty when the agents go overspeeding:

$$r_{\text{speed}} = \begin{cases} -(v_{\text{ego}} - \delta_{\text{velo}}), & v_{\text{ego}} > \delta_{\text{velo}}, \\ 0, & \text{otherwise.} \end{cases}$$

**Constrained Rollout with Pretrained E2E Agents.**   We use the pretrained UniAD [3], VAD [4], and LTF [2] as base behavior policies to generate counterfactual datasets to further train the MPA. We use the planned trajectory for every timestep and augment it with different rotation angles (-5.0° to +5.0°) and scales (0.1 times to 2.0 times), leading to 21 counterfactual behaviors at a single observation

frame. The key parameter during the counterfactual generation is to filter out the undesired samples based on the reward heuristics section discussed in the previous subsection.

For the policy adapter, we exclude all collision samples with a reward of $r_{\text{collision}} = -1$, although these samples are retained in the training dataset of the Q-value network. To ensure reliable learning for both the value network and the policy adapter, it is essential that the 3DGS-rendered datasets exhibit high visual realism. We assess the realism of these renderings using the Fréchet Inception Distance (FID) and Kernel Inception Distance (KID) metrics, as shown in Figure 8. The plot provides insight into acceptable lateral distance thresholds for maintaining high-fidelity simulation. As illustrated in the figure, both FID and KID degrade with increasing lateral displacement. Empirically, a lateral deviation of less than 0.6 m from the closest reference trajectory yields sufficiently realistic renderings, with KID remaining below 0.3 and FID generally under 50.

Figure 8: FID ($\downarrow$) and KID ($\downarrow$) with respect to the lateral distance to the ground truth trajectory.

## A.2 Implementation Details in Training and Inference

**Network Architecture.** The architectures of the diffusion adapter and the value model are illustrated in Figure 9. Both models take the ego history information and action as input. The key difference is that the adapter model takes the base action from the pretrained model, a noisy residual action for the diffusion block, and intermediate representations from the base model, while the value model takes raw camera RGB inputs and final actions to be evaluated and selected at inference time.

**Computing Resources.** The experiments are run on a server with AMD EPYC 7542 32-Core Processor CPU with 256 threads, 4×NVIDIA A5000 graphics, and 252 GB memory. For one experiment, it takes around 6 hours to train around 20 epochs for the policy adapter and value model. At inference time, the inference of the 3DGS-based world model and pretrained base driving model can be run on a single A5000. The inference time of pretrained baselines (UniAD, VAD) is around 0.5s, and the 3DGS world model takes around 0.3s to render all six camera observations with static and dynamic objects in the traffic scenes, and they are running two separate worker nodes following the HUGSIM [13]. For one single scenario, the maximum step is 200. Most scenarios will be finished less than one-third of the maximum time. In total, it will take an average of 1 minute for the evaluation per scene.

**Hyperparameter Table.** We list the hyperparameters used in MPA as follows. They include the hyperparameters for data curation, diffusion adapter, and the value model.

The link to our anonymous codebase is attached at: https://anonymous.4open.science/r/MPA-7432.

## A.3 Additional Description of the Baselines

**UniAD** [3] presents a unified autonomous driving framework that jointly learns perception, prediction, and planning within a single Transformer-based architecture. The method explicitly formulates

Table 4: Hyperparameters for Our Methods.

| Component | Value | Description |
|---|---|---|
| **Policy Adapter Architecture** | | |
| BEV encoder | ResNet-18 | CNN Encoder for BEV input. |
| Ego encoder | 128-dim | Encodes ego history features. |
| Action encoder | 128-dim | Encodes the base actions and noisy target future trajectories. |
| Fused input dimension | 960 | Concatenated input vector. |
| Latent fusion module | 1D U-Net | Applies down/up-sampling over 960-dim vector. |
| Latent dimension | 256 | Feature dimension used throughout latent layers. |
| Residual prediction heads | 1-20 | Each outputs a 12-dimensional trajectory residual. |
| Mixture weight head | 1 | Outputs logits over modes. |
| DDIM steps (training) | 25 | Number of noisy timesteps during training. |
| **Value Network Architecture** | | |
| Image encoder | ResNet-18 | Shared pretrained CNN across 6 stitched views. |
| Image fusion | Mean pooling | Average of per-view features. |
| History steps $T_{\text{hist}}$ | 5 | History length of the ego agents |
| Hidden dimension | 512 | Used throughout encoders and MLPs. |
| Final decoder | MLP | Two-layer predictor for value scalar output. |
| **Training Settings (Policy Adapter)** | | |
| Batch size | 256 | Samples per mini-batch. |
| Learning rate | $1 \times 10^{-4}$ | Initial LR with cosine decay. |
| Optimizer | AdamW | With weight decay of $10^{-4}$. |
| Epochs | 1000 | Total training iterations. |
| Gradient clipping | 1.0 | Global gradient norm threshold. |
| **Training Settings (Value Network)** | | |
| Batch size | 128 | Samples per mini-batch. |
| Learning rate | $1 \times 10^{-4}$ | Optimized via Adam. |
| Epochs | 100 | Total training epochs. |
| Validation ratio | 0.10 | 90% train, 10% validation. |
| Sampler | Weighted | Inverse-frequency sampling for value balance (collision value only). |
| Image input format | 6×(3×224×400) | Stitched from 2×3 grid of 800×450 crops. |
| **Inference-Time** | | |
| DDIM steps (training/sampling) | 5 | Number of diffusion time steps. |
| Noise schedule | Linear | $\beta_t$ in $[10^{-4}, 0.02]$. |
| DDIM $\eta$ | 0.0 | Deterministic sampling. |
| Route Completion Weight $w_{\text{route}}$ | 1000.0 | Penalty weight of route progression |
| Lateral Distance Weight $w_{\text{dist}}$ | 50.0 | Penalty weight of lateral distance away from reference route |
| Collision Value Weight $w_{\text{collision}}$ | 100.0 | Collision penalty weight |
| Speed limit Weight $w_{\text{collision}}$ | 50.0 | Speed penalty weight beyond the limit threshold with *36kph* |

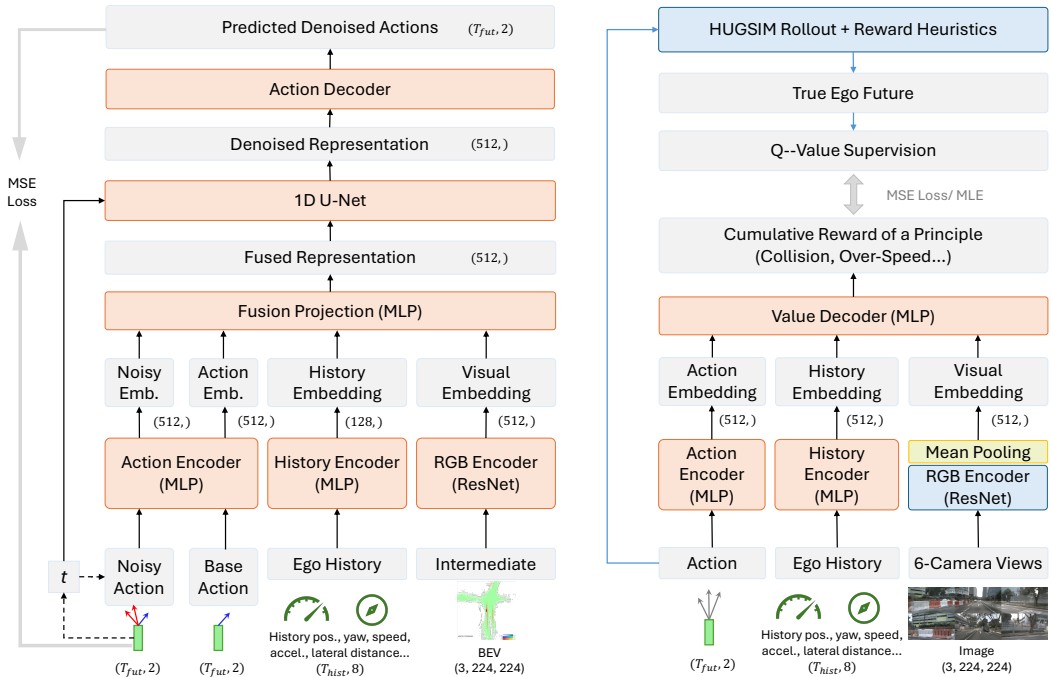

Figure 9: **The Architecture of Policy Adapter and Q-Value Model.** **Left**: The policy adapter consists of the visual encoder, ego history encoder, and action encoder. The embeddings of all three encoders are further fused and passed through a 1D U-Net to get the denoised actions. **Right**: the Q-value model consists of a pretrained visual encoder ResNet-18 with freeze weights. The remaining parts include similar action encoder and history encoder, and each value principle is supervised by the cumulative step-wise heuristic rewards defined during the counterfactual data rollouts in the HUGSIM.

the driving task as an autoregressive trajectory generation problem, leveraging dense Bird's Eye View (BEV) representations to enable strong coordination between scene understanding and motion planning. Through large-scale offline training, UniAD achieves state-of-the-art performance under open-loop evaluation protocols, demonstrating the effectiveness of end-to-end learning for autonomous driving tasks. The codebase in open-sourced at https://github.com/OpenDriveLab/UniAD/ with Apache License 2.0.

**VAD** [4] extends the end-to-end paradigm with a focus on visual grounding and scalable deployment. The framework employs multi-camera inputs to extract spatial-temporal features and utilizes Transformer blocks to capture interactions across different road agents. By removing the reliance on LiDAR sensors or high-definition maps, VAD enables more practical real-world deployment while maintaining high-quality planning capabilities through direct alignment of vision features with future motion targets. The codebase in open-sourced at https://github.com/hustvl/VAD/ with Apache License 2.0.

**Latent TransFuser** [2] represents a camera-only variant of the TransFuser architecture that operates without LiDAR inputs, relying exclusively on multi-view camera imagery. The method fuses features from both perspective and Bird's Eye View representations using cross-attention mechanisms, enabling robust reasoning about scene layout and agent behaviors. Despite the absence of LiDAR data, this approach achieves competitive performance by leveraging latent representations and hierarchical fusion strategies that preserve the spatial structure and temporal dynamics essential for autonomous driving. The codebase in open-sourced at https://github.com/hyzhou404/NAVSIM/ with Apache License 2.0.

**BC-Safe** is implemented by filtering unsafe trajectories and conducting behavior cloning on the E2E driving problem [31]. We adapt the implementation from the safe imitation learning repositories in https://github.com/HenryLHH/fusion/ with the MIT License.

**Diffusion** baseline is implemented by conditioning the action generation on the history trajectory and RGB images directly [32]. We adopt DDIM implementation from the diffusion policy [62] at https://github.com/real-stanford/diffusion_policy with the MIT License.

# B  Additional Experiments Details

In this section, we provide additional experiment results and algorithm implementation details.

## B.1  Additional Environment Description

**Evaluation Metrics.**    Following NAVSIM [22] and HUGSIM [13], we describe all evaluation metrics below:

- **RC**: Route Completion score. It ranges from 0 to 1, representing the progress of the ego vehicle before encountering severe failure such as collision, off-road driving, deviating far from the preset trajectory, or exceeding the time limit.

- **NC**: Non-Collision score. This evaluates the absence of at-fault collisions between the ego agent and surrounding static or dynamic traffic entities. Since the 3DGS model may erode some objects with inconsistent pointclouds, the NC tends to be lower-estimated as some near miss will be counted as collision samples due to the reconstruction error of 3DGS.

- **DAC**: Driveable Area Compliance score. Slightly different from NAVSIM, which has ground truth map information, HUGSIM considers a violation of the driveable area if the ratio of the area within the current ego pose that stays inside the ground point clouds is smaller than 0.3. If this ratio is between 0.3 and 0.5, then the DAC score is 0.5. If the ratio is greater than 0.5, the DAC score is 1.0.

- **TTC**: Time-To-Collision score. If the ego agent fails to pass a collision check within the next 0.5 s with surrounding objects (different from 0.9 s in NAVSIM), the TTC score is 0; otherwise, it is 1.

- **COM**: Comfort score. It is 1 if the longitudinal and lateral acceleration, yaw rate, yaw acceleration, and longitudinal jerk all fall below their respective thresholds; otherwise, it is 0.

- **HDScore**: HUGSIM Driving Score. As introduced in the main text, it is computed based on the above metrics, replacing the Ego Progress in NAVSIM with the Route Completion score:

$$\text{HDScore} = \text{RC} \times \frac{1}{T} \sum_{t=0}^{T} \left\{ \prod_{m \in \{\text{NC, DAC}\}} \text{score}_m \times \frac{\sum_{m \in \{\text{TTC, COM}\}} \text{weight}_m \times \text{score}_m}{\sum_{m \in \{\text{TTC, COM}\}} \text{weight}_m} \right\}_t$$

**Dataset Overview.**    Our environment includes a training split of 290 scenes. The distribution of dynamic objects among the training scenes is illustrated below. Most of the scenes have no more than 40 dynamic traffic entities. Yet this is already sufficient to craft diverse and complex traffic scenes.

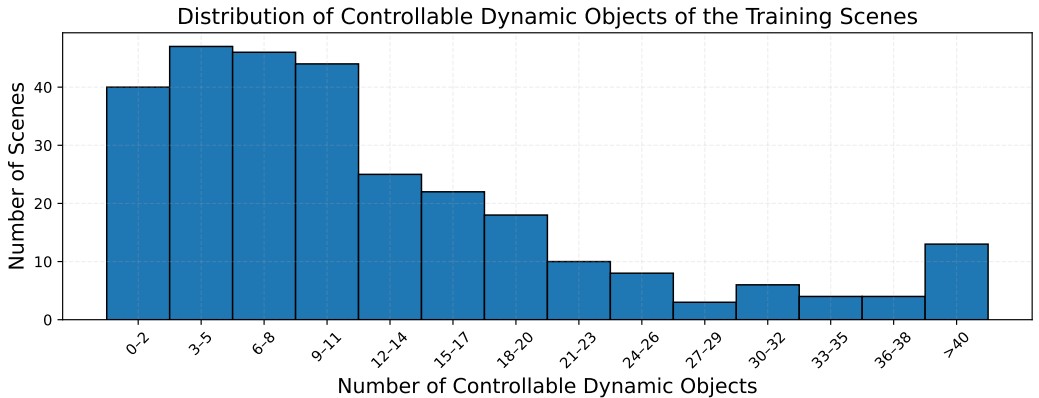

Figure 10: A histogram of the number of dynamic objects in the scenes.

## B.2 Additional Quantitative Results

**Additional Ablation Studies**    As shown in Table 5 and Table 6, we extend the ablation study of MPA to the VAD and LTF base policies. In both settings, the full model with all Q-value guidance and the adapter (ID-6) consistently achieves the highest route completion and HDScore across nominal and safety-critical scenarios. Regarding the adapter, its removal leads to similar performance drops for both VAD and LTF, suggesting that while helpful, the value model plays a more critical role in these cases. Ablating individual Q-components reveals distinct impacts: removing $Q_{\text{route}}$ (ID-1) severely degrades nearly all metrics, especially RC and TTC; removing $Q_{\text{dist}}$ (ID-2) significantly harms DAC and HDScore; excluding $Q_{\text{collision}}$ (ID-3) causes agents to terminate earlier in safety-critical scenes due to collisions, lowering RC and HDScore; and omitting $Q_{\text{speed}}$ (ID-4) reduces COM and slightly affects TTC, leading to a modest decline in HDScore. These findings about value guidance modules align with our results on the UniAD policy and further validate the generalizability of our MPA design.

Table 5: **Ablation Study** on MPA's variants on the VAD base policy. **Top**: Unseen scenes that are nominal but not appearing in the training dataset. **Bottom**: Safety-critical scenes with adversarial surrounding agents. **Bold** means the best, and underlined is the best runner-up for each metrics.

| ID | $Q_{\text{route}}$ | $Q_{\text{dist}}$ | $Q_{\text{collision}}$ | $Q_{\text{speed}}$ | Adapter | RC | NC | DAC | TTC | COM | HDScore |
|----|------|------|------|------|------|------|------|------|------|------|------|
| 1 |  | ✓ | ✓ | ✓ |  | 10.2 | 77.2 | 94.5 | 77.1 | 100.0 | 7.6 |
| 2 | ✓ |  | ✓ | ✓ |  | 82.2 | 59.5 | 86.1 | 57.5 | 97.2 | 47.7 |
| 3 | ✓ | ✓ |  | ✓ |  | 90.6 | 70.5 | 94.5 | 69.0 | 97.7 | 60.7 |
| 4 | ✓ | ✓ | ✓ |  |  | 92.2 | 71.4 | 94.0 | 68.9 | 97.6 | 61.9 |
| 5 | ✓ | ✓ | ✓ | ✓ |  | 88.9 | 73.6 | 94.8 | 71.2 | 97.8 | 62.6 |
| 6 | ✓ | ✓ | ✓ | ✓ | ✓ | 90.9 | 71.0 | 94.4 | 68.8 | 97.7 | 61.2 |

| ID | $Q_{\text{route}}$ | $Q_{\text{dist}}$ | $Q_{\text{collision}}$ | $Q_{\text{speed}}$ | Adapter | RC | NC | DAC | TTC | COM | HDScore |
|----|------|------|------|------|------|------|------|------|------|------|------|
| 1 |  | ✓ | ✓ | ✓ |  | 14.4 | 89.2 | 99.3 | 84.8 | 96.4 | 13.1 |
| 2 | ✓ |  | ✓ | ✓ |  | 76.9 | 72.3 | 90.6 | 60.5 | 99.3 | 52.8 |
| 3 | ✓ | ✓ |  | ✓ |  | 76.1 | 78.4 | 99.8 | 68.6 | 97.7 | 55.0 |
| 4 | ✓ | ✓ | ✓ |  |  | 98.2 | 80.9 | 99.8 | 70.9 | 99.3 | 71.8 |
| 5 | ✓ | ✓ | ✓ | ✓ |  | 98.3 | 81.6 | 99.4 | 71.3 | 99.3 | 72.2 |
| 6 | ✓ | ✓ | ✓ | ✓ | ✓ | 96.6 | 79.8 | 99.0 | 77.3 | 97.7 | 74.7 |

Table 6: **Ablation Study** on MPA's variants on the LTF base policy. **Top**: Unseen scenes that are nominal but not appearing in the training dataset. **Bottom**: Safety-critical scenes with adversarial surrounding agents. **Bold** means the best, and underlined is the best runner-up for each metrics.

| ID | $Q_{\text{route}}$ | $Q_{\text{dist}}$ | $Q_{\text{collision}}$ | $Q_{\text{speed}}$ | Adapter | RC | NC | DAC | TTC | COM | HDScore |
|----|------|------|------|------|------|------|------|------|------|------|------|
| 1 |  | ✓ | ✓ | ✓ |  | 11.0 | 80.0 | 95.5 | 79.8 | 94.9 | 7.9 |
| 2 | ✓ |  | ✓ | ✓ |  | 85.3 | 63.4 | 83.0 | 58.9 | 97.2 | 48.3 |
| 3 | ✓ | ✓ |  | ✓ |  | 89.6 | 67.6 | 90.9 | 65.3 | 96.9 | 55.4 |
| 4 | ✓ | ✓ | ✓ |  |  | 91.1 | 66.4 | 84.7 | 63.1 | 96.6 | 49.8 |
| 5 | ✓ | ✓ | ✓ | ✓ |  | 89.8 | 69.6 | 91.7 | 64.7 | 94.7 | 57.9 |
| 6 | ✓ | ✓ | ✓ | ✓ | ✓ | 91.8 | 68.3 | 91.0 | 66.5 | 96.9 | 57.0 |

| ID | $Q_{\text{route}}$ | $Q_{\text{dist}}$ | $Q_{\text{collision}}$ | $Q_{\text{speed}}$ | Adapter | RC | NC | DAC | TTC | COM | HDScore |
|----|------|------|------|------|------|------|------|------|------|------|------|
| 1 |  | ✓ | ✓ | ✓ |  | 7.2 | 94.1 | 100.0 | 92.8 | 67.5 | 6.2 |
| 2 | ✓ |  | ✓ | ✓ |  | 63.4 | 74.2 | 91.4 | 63.5 | 99.2 | 36.5 |
| 3 | ✓ | ✓ |  | ✓ |  | 59.4 | 76.1 | 98.6 | 61.4 | 98.9 | 39.3 |
| 4 | ✓ | ✓ | ✓ |  |  | 88.7 | 74.9 | 93.9 | 64.6 | 98.3 | 54.7 |
| 5 | ✓ | ✓ | ✓ | ✓ |  | 89.9 | 76.9 | 97.8 | 70.1 | 98.7 | 61.7 |
| 6 | ✓ | ✓ | ✓ | ✓ | ✓ | 87.3 | 72.0 | 94.0 | 66.9 | 97.8 | 56.3 |

**Impact of the Size of Adapter Modalities**    We evaluate the performance of MPA models over the safety-critical testing environments. The results show that larger sizes of modalities consistently bring benefits to the driving performance in RC, DAC, and HDScore, especially under the smaller mode sizes from 1 to 8. Properly adding the number of modes benefits the closed-loop driving performance.

**Statistical Significance Analysis.**    Table 7 demonstrates 3-seed over the safety-critical scenes. The results show a reasonably consistent performance of all the methods with low standard deviation

across different random seeds. Our MPA is still significantly better than the baselines among all the safety-critical scenes.

Table 7: **Statistical Significance Results** in unseen safety-critical scenes. We illustrate the mean and standard deviation of all the metrics tested with three random seeds for all the baseline methods and our approach. All the evaluation metrics are higher the better. **Bold** means the best, and underlined is the best runner-up for each metrics. Based on the mean and standard deviation results among three random seeds, MPA-empowered methods can significantly outperform baselines in RC and HDScore under the safety-critical scenes.

| Model | RC | NC | DAC | TTC | COM | HDScore |
|---|---|---|---|---|---|---|
| UniAD | $11.4_{\pm0.05}$ | $76.9_{\pm0.62}$ | $81.8_{\pm0.38}$ | $58.3_{\pm0.39}$ | $83.0_{\pm9.15}$ | $4.3_{\pm0.17}$ |
| VAD | $23.5_{\pm4.77}$ | $76.5_{\pm0.39}$ | $88.3_{\pm0.00}$ | $69.7_{\pm3.52}$ | $96.1_{\pm5.47}$ | $12.9_{\pm3.36}$ |
| LTF | $34.0_{\pm0.98}$ | $81.0_{\pm0.29}$ | $96.1_{\pm0.71}$ | $72.3_{\pm4.21}$ | $99.4_{\pm0.46}$ | $22.0_{\pm1.69}$ |
| AD-MLP | $4.9_{\pm0.33}$ | $93.0_{\pm0.68}$ | $96.9_{\pm0.67}$ | $91.3_{\pm1.71}$ | $70.4_{\pm11.39}$ | $4.1_{\pm0.39}$ |
| BC-Safe | $20.8_{\pm0.76}$ | $79.3_{\pm0.70}$ | $91.7_{\pm0.65}$ | $65.4_{\pm1.39}$ | $95.6_{\pm6.27}$ | $13.1_{\pm0.33}$ |
| Diffusion | $26.5_{\pm5.04}$ | $87.1_{\pm2.17}$ | $95.7_{\pm2.46}$ | $72.6_{\pm0.78}$ | $92.6_{\pm4.48}$ | $16.7_{\pm3.59}$ |
| MPA (UniAD) | $89.1_{\pm4.29}$ | $79.6_{\pm2.68}$ | $99.1_{\pm0.17}$ | $70.6_{\pm2.64}$ | $97.2_{\pm0.62}$ | $63.9_{\pm4.63}$ |
| MPA (VAD) | $97.8_{\pm0.83}$ | $81.8_{\pm1.76}$ | $99.2_{\pm0.17}$ | $73.8_{\pm2.54}$ | $98.8_{\pm0.75}$ | $73.6_{\pm1.04}$ |
| MPA (LTF) | $89.2_{\pm1.33}$ | $73.4_{\pm2.46}$ | $96.7_{\pm1.89}$ | $67.3_{\pm2.14}$ | $97.5_{\pm1.17}$ | $58.5_{\pm2.30}$ |

## B.3 Additional Qualitative Studies

We illustrate more qualitative results in both counterfactual data generation and closed-loop evaluation of different E2E driving agents.

**More Counterfactual Data Generation Examples.** As illustrated in Figure 11, we illustrate some additional counterfactual data generation examples on six scenarios of the training split. The results show that the augmented data improves the data coverage, leading to more robust policy and value learning in MPA.

**Comparison of Driving Performance of In-Domain and Safety-Critical Scenes.** We further visualize the performance of MPA and baselines in nominal and safety-critical scenarios in Figure 12 and Figure 13. We can observe that the baselines without the value model usually fail to meet the safe constraints, leading to catastrophic consequences such as collision or getting off-road.

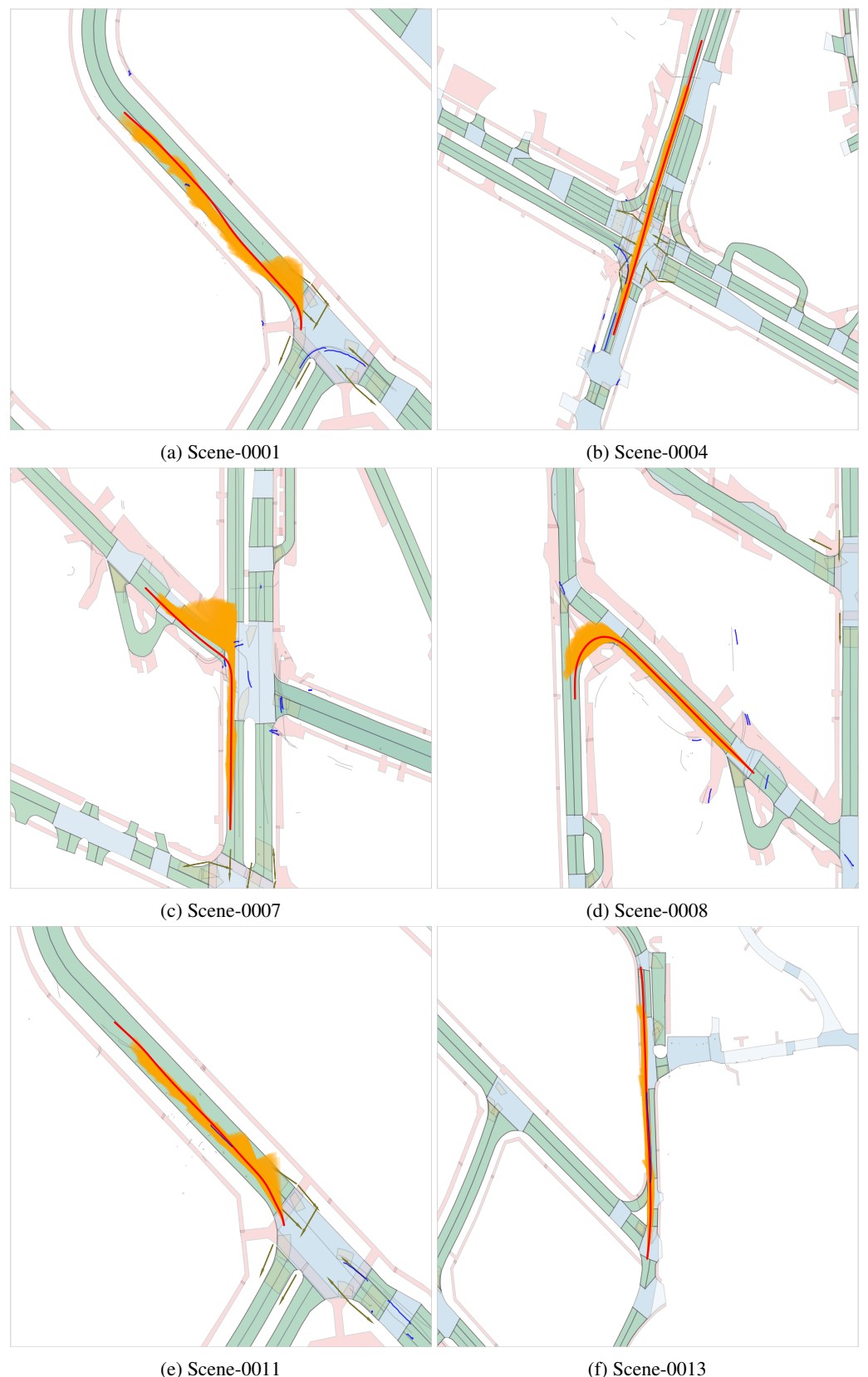

(a) Scene-0001

(b) Scene-0004

(c) Scene-0007

(d) Scene-0008

(e) Scene-0011

(f) Scene-0013

Figure 11: **Examples of Counterfactual Data Generation**. **Red lines** represent the ground truth trajectory. **Orange lines** represent the augmented counterfactual trajectories that are used to train Q-value model and policy adapter. **Blue lines** represent the other agents' behavior at the scene.

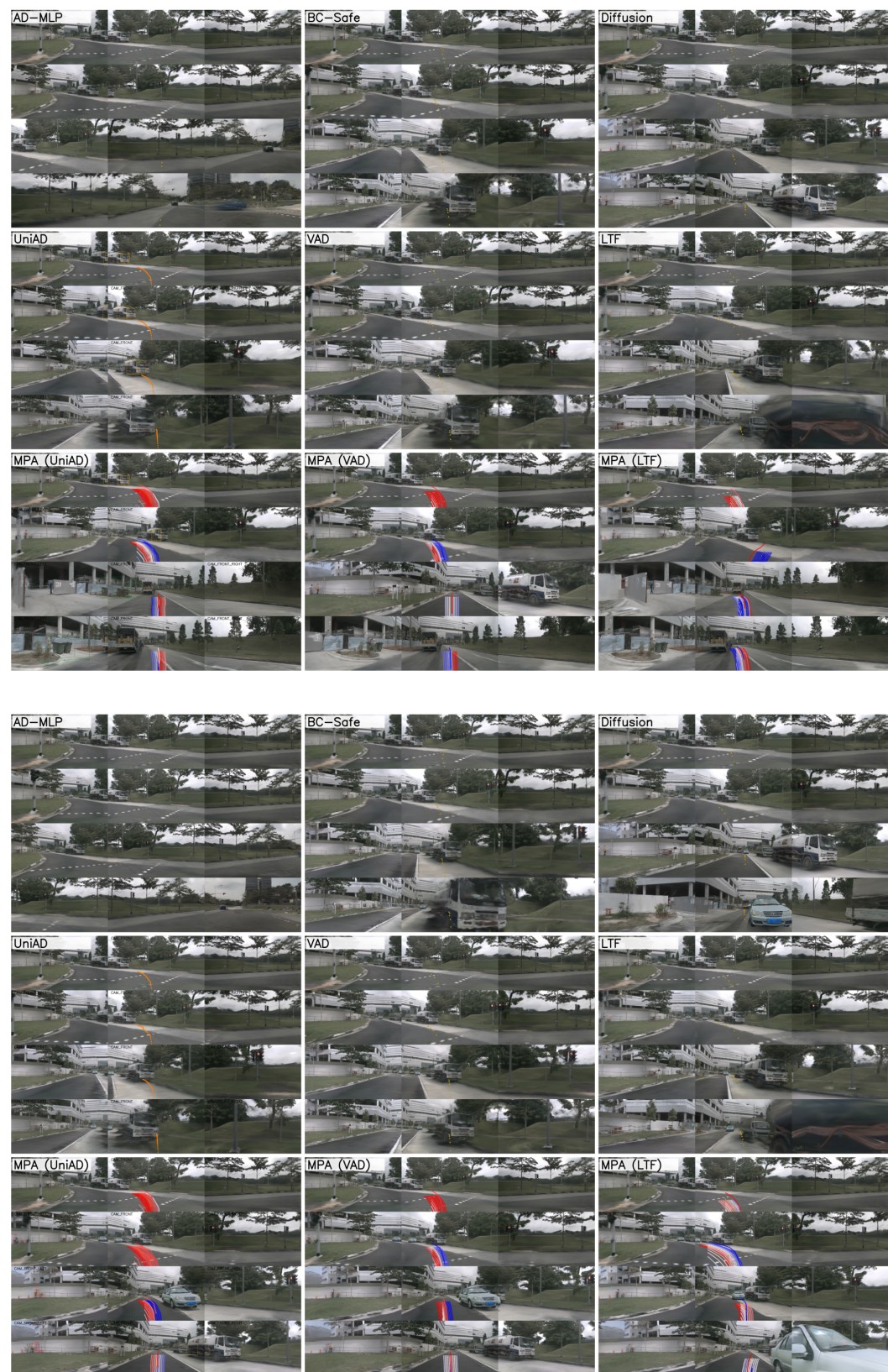

Figure 12: **Qualitative Results** of Scene-0001. **Top**: In-distribution scenes where ego agents turn left and yield to pedestrians. **Bottom**: Safety-critical scenes, ego agents turn left and encounter a fast-approaching oncoming vehicle. MPA agents take high value actions rather than low value ones.

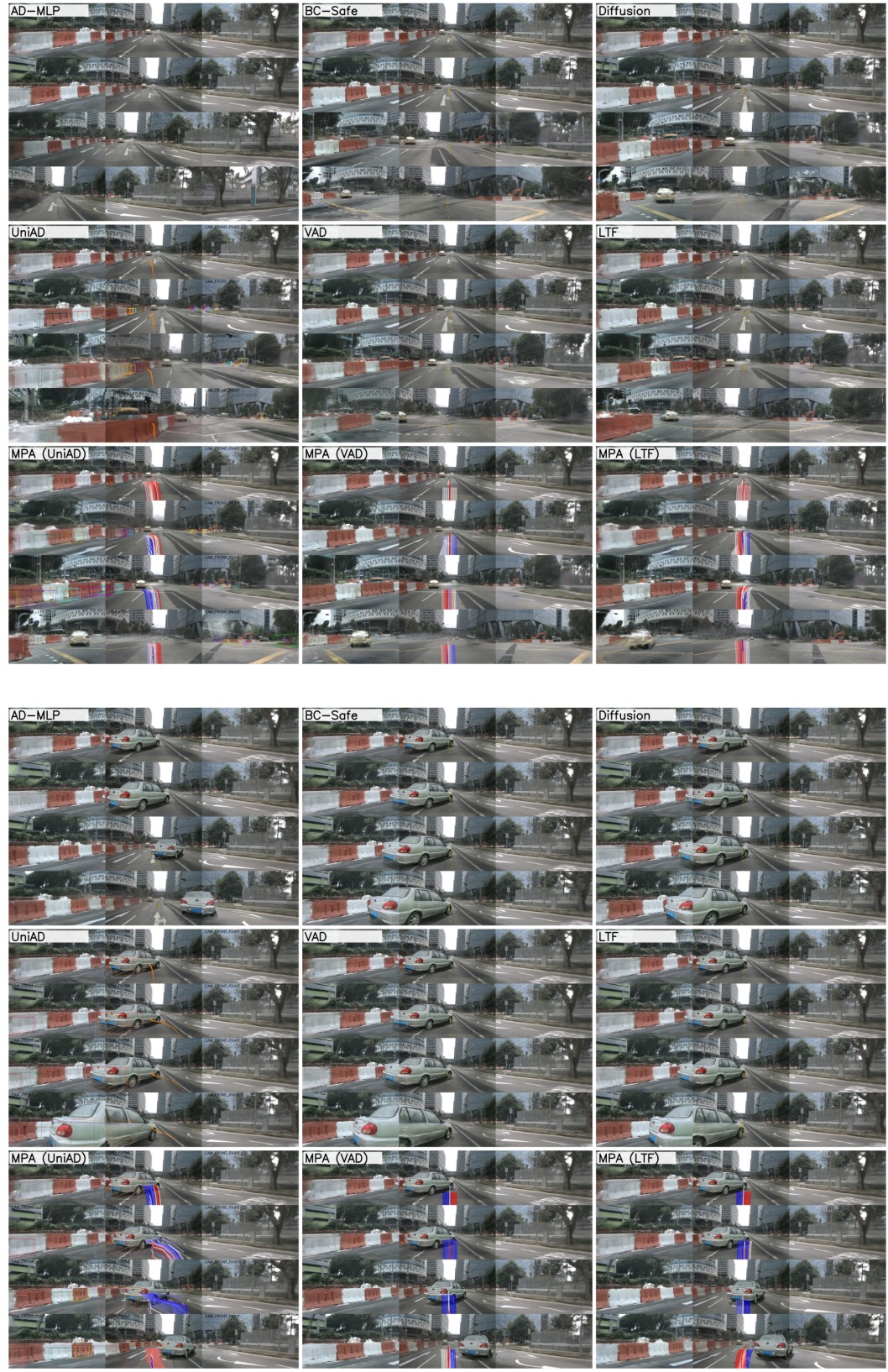

Figure 13: **Qualitative Results** of Scene-0004. **Top**: In-distribution scenes where ego agents move forward and cross the intersection. **Bottom**: Safety-critical scenes with an agent cutting in from the left lane, ego agents need to brake and yield to the cut-in agent. MPA agents take high value actions instead of low value ones.

# C  Broader Impact

The proposed Model-Based Policy Adaptation (MPA) framework enhances end-to-end autonomous driving by improving safety and robustness in complex scenarios through high-fidelity simulation and reward-guided inference. This can accelerate deployment and reduce accidents. However, risks include biased reward models, gaps between simulated and real-world performance, potential misuse for surveillance or adversarial purposes, and privacy concerns from real-world data. Mitigation strategies include gated model release, reward interpretability, simulation validation, and data anonymization.

