# OpenReview forum: "Model-Based Policy Adaptation for Closed-Loop End-to-end Autonomous Driving"
_NeurIPS.cc/2025/Conference — NeurIPS 2025 poster_

### Official Review · Reviewer_7qBw · 2025-06-27

**Clarity:** 3
**Significance:** 4
**Originality:** 4
**Rating:** 5
**Confidence:** 3

**Summary:**

The paper presents Model-based Policy Adaptation (MPA), a framework that improves the robustness of end-to-end driving policies in closed-loop settings. The submission first analyses issues of such driving models that occur under distribution shifts of closed-loop simulation. Moreover, a data generation pipeline is proposed that unrolls a reference policy on reference trajectories in a simulator based on 3D Gaussian Splatting (3DGS) to generate novel observations. This counterfactual data is utilized to (1) train a diffusion-based policy adapter for the trajectories and (2) an action-value model that guides the adapter during inference. Experiments using the HUGSIM simulation tools and the nuScenes dataset demonstrate the benefits of the proposed robustness strategy with several reference policies and ablate relevant design choices.

**Questions:**

- Could you comment on “…increasing the planning frequency and shortening the planning horizon cannot necessarily close the gap between open-loop and closed-loop performance, unless some effective feedback guidance is provided to the E2E agents.” in Section 2.2? I am not sure if that is a clear consequence from the analysis.
- Are the reference trajectories simply the ground truth states of the ego vehicle, sampled at different time steps in the scene? Using the word “reference” for both the policy and the recorded dataset might not be ideal.
- Could you provide some intuition or qualitative result on why LTF benefits less from the robustness strategy compared to UniAD and VAD (despite being more robust without MPA)? Is this related to the smaller intermediate representation?
- Could you comment on the additional compute requirements of MPA (i.e. with the diffusion adapter and value prediction)? Do these modules substantially increase run-time?

I am quite positive about this work. I am open to increasing my overall score, if unclarities and questions can be sufficiently resolved.

**Ethical Concerns:**

["NO or VERY MINOR ethics concerns only"]

**Final Justification:**

I recommend to accept this paper. The topic is highly relevant, and the paper has a good evaluation setup and strong results.

**Limitations:**

The submission discusses limitations. The authors might want to consider commenting on the compute requirements, as such methods would need to be very efficient in a real-world setting. Moreover, I personally think that long simulations (i.e. in the synthetic CARLA simulator) might be required to support some claims regarding long-term planning benefits with the Q value model. Though, I respect that this is beyond the scope of this project.

**Paper Formatting Concerns:**

- Line 71: Is it supposed to mean “reward function”?
- Line 114: There is some issue in the line. Either a point is missing or the word “Even” was not intended here.
- Line 956 (supplementary pdf): Missing figure reference.

**Quality:**

3

**Strengths And Weaknesses:**

Strengths:
- The topic is highly relevant and well motivated in the submission. With end-to-end driving models being more and more used in industry and academia, addressing the closed-loop robustness is a promising field. The proposed framework does not pose compatibility constraints to the specific driving policy. The work appears highly valuable to the community due the quality of the work and tools that are likely open-sourced.
- The paper is mostly well written. Specifically the experimental study is well structured and clear to follow. It is apparent from the paper and supplementary pdf that thorough work was required for the project.
- The experiments are conducted with closed-loop simulation based on state-of-the-art sensor simulation (i.e. 3DGS).  The claims are well supported. The authors discuss limitations of the approach.

Weaknesses:
- My main issue with the submission is that I do not find the counterfactual data generation procedure clearly described. This negatively impacts the understanding of the downstream diffusion adapter and value learning modules. I find it somewhat unclear how the reference trajectories are constructed, how the reference policy is applied (and on which time steps), and how the reward is calculated. A figure or more precise notation might be helpful. For example, Figure 3 provides a BEV overview where the colors are not defined, the trajectories are not clearly visible, and the procedure is not shown over the different timesteps.
- I am uncertain how much value Section 2.2 (and partially Section 2.1) provides in the paper. It is well known that behavioral cloning suffers from distributional shifts and compounding errors, thus the section does not provide much value to the reader. I recommend shortening these parts or providing further insights or qualitative examples instead.

---

> ### Author Rebuttal · Authors · 2025-07-31
>
> We sincerely thank reviewer 7qBw for their insightful and inspiring feedback. We are glad to know the reviewer recognizes the motivation and community value of our work, the generality of the MPA framework, as well as its strong empirical performance. We provide our response to the questions below.
>
> **W1. Clarification on the counterfactual data generation procedure.**
>
> We thank the reviewer for raising this important clarification question. We will make clarification on each question as follows.
>
>
> * **W1.1 (Q2) How are the reference trajectories constructed? Are they simply the ground truth states of the ego vehicle? How the reference policy is applied (and on which time steps)?**
>
> The reference policy is not the ground truth states of ego vehicle. Instead, they are constructed by adding random noise to the base policy's prediction (UniAD, VAD, LTF). In practice, we conduct geometric transformation by rotating the trajectories with a random angles uniformly sampled from -5 degree to 5 degree, and also scaled the longitude motions with random scales from 0.1 to 2.0.
>
> The the counterfactual data generated by this reference policy consists of over 20 counterfactual actions under the same states, and their reward will be annotated by the underlying states from HUGSIM during training time. Similar techniques are adopted by the PDM controller in [1].
>
> Note that this 'reference policy' is only applied during training time to help collect counterfactual datasets. We will modify the term 'reference policy' to 'behavior policy' in our revised manuscript to avoid further misunderstanding.
>
>
> * **W1.2 How is the reward calculated?**
>
> The reward is calculated by the underlying states from HUGSIM simulator. It is a weighted sum of route completion, lateral distance, speed, and the collision penalty. We attach a more detailed description of the reward definition in Appendix Section A.1.
>
>
> * **W1.3 A figure or more precise notation might be helpful. For example, Figure 3 provides a BEV overview where the colors are not defined, the trajectories are not clearly visible, and the procedure is not shown over the different timesteps.**
>
> We thank the reviewer's suggestion in our visualization and will revise it to be more clear in our revised manuscripts. Currently, the BEV map we used is the identical as the direct BEV map prediction from our base policies, and the one in Figure 3 is the visualization from UniAD. For every step of rollouts, the BEV map will be updated, and the model will take in the most recent observations to correct the actions. We will add a multi-step demonstration to clearly demonstrate the inference-time pipeline of MPA.
>
>
> **W2. Regarding reorganizing on section 2.1 and 2.2.**
>
>
> We thank the reviewer for giving this valuable suggestion on our paper writing. While compounding errors is widely acknowledged in RL community, prior end-to-end driving methods fail to resolve it due to the lack of high-fidelity counterfactual data generation and robust policy adaptation algorithm. We will shorten the section in our revision.
>
>
>
> **Q1. Clarification on the claim: "…increasing the planning frequency and shortening the planning horizon cannot necessarily close the gap between open-loop and closed-loop performance, unless some effective feedback guidance is provided to the E2E agents."**
>
> We thank the reviewer for the careful reading and thoughtful feedback on our presentation. We mainly draw this claim from Figure 2, where the closed-loop trajectory prediction yields to compounding error as the planning horizon enlarges. We also notice from Figure 2 that the L2 prediction error is non-neglegible **even under small prediction horizon** (0.5s\~1s). This suggests that **behavior cloning loss is not a strong supervision signal** even for a short-horizon policy. Then we propose to use other guidance signal (such as reward in our counterfactual dataset) to help correcting the prediction online. We will add more elaboration between the analysis on Figure 2 and this claim to make the presentation more coherent.
>
> **Q3. Qualitative studies on the LTF's performance gain from MPA.**
>
> We thank the reviewers for their thoughtful feedback. Since no images are allowed in the response, we cannot visualize the heatmap of predicted trajectories, but we'll add them in our revised version. **Alternatively**, in the tables below, we analyze the statistics in the driving behaviors of LTF, UniAD, VAD, and MPA correction effect on them.
> * We quantify the diversity in both **longitude** and **lateral** directions, for both in-distribution and unseen scenarios. We can see a clear difference in the longitude prediction's standard deviation between LTF and other methods without MPA. The metrics are reported with the future waypoints in 3 seconds with a unit of meter. In both tables, LTF base model is **more agile with higher standard deviation** compared to UniAD and VAD in the longitude direction, which demonstrates their capability to accelerate/decelerate to avoid collision or getting into other safety-critical conditions.
> * We also notice for all the methods, after MPA's correction, their standard deviation in longitude (and most lateral, except for VAD in OOD scenes) predictions will increase. This partly reveals some internal mechanism in how the MPA improves the behavior quality: it gives the base model **better agility** in both longitude and lateral movements.
> * Although MPA benefits LTF, the beneficial effect is not as large as the others. Since we use the same random trajectory transformation during the counterfactual data generation perspective, this may lead to worse coverage in LTF in the whole trajectory space, since the base action in LTF is already quite diversified.
>
> **Table R1.1 Standard deviation in in-distribution scenarios. Without MPA, LTF already has higher longitude behavior diversity compared to the other base policies.**
> |Method|Longitude (w/o MPA)|Longitude (w/ MPA)|Lateral (w/o MPA)|Lateral (w/ MPA)|
> |-|-|-|-|-|
> |UniAD|2.78|9.46|1.10|1.53|
> |VAD|1.53|9.16|0.82|1.28|
> |LTF|**8.77**|11.91|1.12|1.49|
>
> **Table R1.2 Standard deviation unseen scenarios, Without MPA, LTF already has higher longitude behavior diversity compared to the other base policies.**
> |Method|Longitude (w/o MPA)|Longitude (w/ MPA)|Lateral (w/o MPA)|Lateral (w/ MPA)|
> |-|-|-|-|-|
> |UniAD|2.64|10.74|0.99|1.17|
> |VAD|1.45|10.35|1.53|1.41|
> |LTF|**9.52**|12.86|1.06|1.32|
>
>
> We also quantify the distribution of output action corrections $\Delta \hat{a} \approx a^*-a_{base}$. In the following table, we quantify the standard deviation of lateral behavior correction from MPA in both in-distribution (ID) and unseen (OOD) scenes. MPA(LTF) has the highest deviation in lateral correction, which leads to more unstable steering actions. This potentially reduces the final performance in driveable area compliance (related to DAC) and distance keeping (related to NC and TTC) compared to MPA (UniAD) and MPA (VAD).
>
> **Table R2. MPA Correction Analysis. MPA (LTF) has higher lateral deviation compared to the other two methods.**
> |Method|Lateral Correction (ID)|Lateral Correction (OOD)|Steering (ID)|Steering (OOD)|
> |-|-|-|-|-|
> |MPA (UniAD)|1.00|1.02|0.12|0.12|
> |MPA (VAD)|0.71|1.12|0.14|0.18|
> |MPA (LTF)|**1.32**|**1.42**|**0.18**|**0.19**|
>
> **Regarding compact representation**: Given our current empirical results, there is no clear evidence that the intermediate representation of LTF has significant harm in the final performance of MPA. This intermediate representation will be fed to the policy adapter, while the Q-value models have similar capacity as they take in the raw camera observations.
>
>
> **Q4. Discussion on the computation efficiency.**
>
> We ran our experiment on 2xNVIDIA RTX A5000 GPUs for 1000 rollout steps, and calculate the average time comsumption over different modules in MPA:
>
> **Table R3. Inference overhead analysis on MPA.**
> |Component | Time(s) | #Params |
> |-|-|-|
> |Base Policy (UniAD) | 0.67 | 131.81M |
> |Base Policy (VAD)| 0.34  | 58.36M |
> |Base Policy (LTF)| 0.33  | 56.04M |
> |Diffusion Adapter | 0.09 | 12.26M |
> |Q Value Net | 0.07 | 13.84M |
>
> Compared to the inference speed of base policy, the overhead of the additional parameters from MPA is small (a total of 0.16s, <25\% inference time overhead for UniAD, <50\% for VAD and LTF), while boosting up the final performance significantly given a considerably large amount of counterfactual samples and multi-step Q value feedback. We will explore optimizing more efficient (parallel) rollout for MPA in the future.
>
> > [1] Dauner, Daniel, et al. "Parting with misconceptions about learning-based vehicle motion planning." Conference on Robot Learning. PMLR, 2023.

---

> > ### Comment · Reviewer_7qBw · 2025-08-07
> >
> > I thank the authors for addressing my concerns and answering my questions. I encourage the authors to revise the manuscript, i.e. improve the clarity of the counterfactual data generation (W1), adding qualitative examples of performance gains (Q3), and discussing the computation efficiency (Q4). The paper is stronger for these changes, and I maintain my view that this represents a meaningful contribution to the field.

---

> > > ### Author Response · Authors · 2025-08-08
> > >
> > > We appreciate the dedicated efforts and the insightful reviews from reviewer 7qBw. The contributive review helps improve our paper's quality during the revision and rebuttal phase. We will update our revised manuscript accordingly.

---

### Official Review · Reviewer_2ZCm · 2025-07-02

**Clarity:** 2
**Significance:** 3
**Originality:** 3
**Rating:** 5
**Confidence:** 3

**Summary:**

This paper addresses the performance degradation of end-to-end (E2E) autonomous driving models when transitioning from open-loop to closed-loop evaluation. The authors identify a fundamental mismatch is the objective mismatch due to the absence of reward feedback during offline imitation learning. To address that, they propose Model-Based Policy Adaptation (MPA), which consists of three components: counterfactual data generation using 3D Gaussian Splatting (3DGS) simulation, a diffusion-based policy adapter that predicts residual actions, and a multi-step Q-value model for inference-time action selection. The models are trained on nuScenes data and evaluated via HUGSIM simulation engine. Results show that the proposed MPA can outperform the baseline by a large margin, especially in the safety-critical scenes.

**Questions:**

1. In the qualitative results in Fig.4, it seems that MPA is not yielding/breaking compared to the VAD baseline according to the simulated surroundings (the position of road mark / pole, etc.) unless the images are not "simulated on the same timestamp". Could you be more specific about how you obtained the qualitative results?
2. According to the original HUGSIM paper, UniSIM generally performs better than VAD on HDScore, which is contradictory to the authors results. Could you explain why it is the case?
3. The weighted sum of the value model seems ad-hoc to me, could the authors explain how did these weight coefficient to be determined?
4. What is the computation overhead for adding the value model and adapter during inference?

**Ethical Concerns:**

["NO or VERY MINOR ethics concerns only"]

**Final Justification:**

I appreciate the authors' detailed response and decide to keep my score as accept.

**Quality:**

3

**Strengths And Weaknesses:**

Strengths:
1. While individual components are not novel, their combination for addressing the open-loop to closed-loop gap in E2E driving is original, particularly the use of counterfactual data generation with 3DGS simulation.
2. The experimental evaluation is thorough, covering in-domain, out-of-domain, and safety-critical scenarios with consistent improvements across multiple base policies (UniAD, VAD, LTF).
3. The overall methodology is clearly presented with good visual aids (Figures 1 and 3 effectively illustrate the approach).
4. The work provides immediate practical value by showing how to improve existing E2E driving models' deployment safety, with HDScore improvements by a large margin.

Weaknesses:
1. The counterfactual data generation relies heavily on teacher-forcing, which may not explore sufficient behavioral diversity to handle truly out-of-distribution scenarios.
2. The claim that observation mismatch is minor lacks convincing evidence—Figure 2 only addresses motion prediction, not visual fidelity. 3. The Q-value decomposition into four specific components appears arbitrary without ablation on alternative decompositions or theoretical justification.
4. Computational costs are not discussed.
5. For safety-critical scenes, only 10 scenes are used for evaluation, whether this is representative is questionable.

---

> ### Author Rebuttal · Authors · 2025-07-31
>
> We thank Reviewer 2ZCm for the thoughtful and constructive feedback, and for acknowledging the clarity, practical value, and strong empirical performance of our MPA framework in bridging open-loop and closed-loop end-to-end driving. We address each question as follows.
>
> **W1.The counterfactual data generation relies heavily on teacher-forcing, which may not explore sufficient behavioral diversity to handle truly out-of-distribution scenarios.**
>
> We thank the reviewer for this insightful question. The key goal in our counterfactual generation is to **balance the diversity of the sampled trajectories and the fidelity of the rendered images.**
>
> * For the **diversity** of the data, we add random noise by scaling and rotating the predicted trajectories from the base policy. In this way, we can collect the state and action transition pairs under different future motions in the next few frames. We also conduct an ablation study on the counterfactual rollout steps in Figure 5 in our original manuscript. Longer counterfactual rollout steps bring in better diversity in the dataset. We can see that the performance of MPA increases significantly from step 0 to 3, and then converges around 4 and 5 steps. This suggests the current behavioral diversity in the counterfactual dataset is sufficient to train robust policy adapters and Q value models.
> * Meanwhile, for the sake of rendering **fidelity**, we need to constrain the divergence of the sampled counterfactual trajectories. As visualized in Figure 6 in our supplementary material, the quality of the 3DGS-rendered images degrades (FID gets larger than 50) as the lateral distance to the ground truth trajectories gets too large. To avoid this, we will truncate the samples that violates the distance or reward threshold during the counterfactual data generation phase, as included in Algorithm 1 in our main text.
>
> **W2. About the visual fidelity to support the observation mismatch claim.**
>
> HUGSIM allows us to script diverse, photorealistic behaviors for dynamic objects, enabling us to synthesize realistic safety-critical interactions and mitigate the scarcity of real-world interaction-rich data.
>
> As shown in Fig. 6 of the supplementary material, HUGSIM renders high-fidelity images when the synthesized viewpoint remains close to the ground-truth path (FID ~20). Visual quality degrades only as the lateral deviation grows: once the offset exceeds ~0.8 m the FID can surpass 50. Such large deviations occasionally arise when the base policies (UniAD, VAD, LTF) drift during closed-loop rollouts.
>
> Accordingly, when producing counterfactual data, we will constrain the lateral displacement, ensuring every rendered observation remains photorealistic and avoids perception noise that could degrade MPA's final performance.
>
>
>
> **W3 & Q3. Further explanation on the Q-value decomposition into the specific components, as well as the weighted sum of different terms.**
>
> We follow the prior reward recipes in [1] to decompose the essential value function terms in autonomous driving. Note that some of these terms are under different scales in our case. For instance, the route completion is 0 to 1 across all timesteps, and the difference between different timesteps will be very small. Distance to the driving lane has a unit of meters (m). The speed is m/s, and collision is a binary term of 0 or 1.
>
> In order to select the value coefficients, we zoom into the empirical distribution of these individual reward term among the training data, then balancing them by timing them approximately by the reciprocol of their mean, i.e. $w_\text{collision} \approx 1/\text{mean}(r_\text{collision})$, forming the final weights coefficients as presented in the hyparameter table in appendix Table 4.
>
> **W4 & Q4. Computational costs are not discussed.**
>
> We thank the reviewer for raising these valuable points. We ran our experiment on 2xNVIDIA RTX A5000 GPUs for 1000 rollout steps, and report the average time consumption among different modules in MPA:
>
> **Table R1. Inference overhead analysis on MPA.**
> |Component|Time(s)|#Params|
> |-|-|-|
> |Base Policy (UniAD)|0.67|131.81M|
> |Base Policy (VAD)|0.34|58.36M|
> |Base Policy (LTF)|0.33|56.04M|
> |Diffusion Adapter|0.09|12.26M|
> |Q Value Net|0.07|13.84M|
>
> Compared to the inference speed of base policy, the overhead of the additional parameters from MPA is small (a total of 0.16s, <25\% inference time overhead for UniAD, <50\% for VAD and LTF), while boosting up the final performance significantly given a considerably large amount of counterfactual samples and multi-step Q value feedback. We will explore optimizing more efficient (parallel) rollout for MPA in the future.
>
>
> **W5. Limited number of safety-critical scenes.**
>
> We thank the reviewer for highlighting this concern. Generating diverse, safety-critical scenarios remains challenging because real-world crash data are scarce and reliable ego policies for automated stress-testing are still under development. Even without explicitly adversarial scenes, we find that state-of-the-art E2E policies (UniAD, VAD, LTF) frequently fail in **closed-loop rollouts**. Since the main focus of this paper is to first train a robust end-to-end policy, we manually design 10 safety-critical scenarios in the scenes in Singapore from nuScenes dataset.
>
> The current ten safety-critical scenes we curated from HUGSIM also align with other recent benchmarks such as NeuroNCAP [2], where they evaluate the ego policy in five front scenes and five side scenes from the nuScenes dataset.
>
> In the future, we intend to use MPA as a robust ego policy (i.e. "victim policy") that is ready for a more scalable safety-critical scenario generation pipeline in the end-to-end autonomous driving research.
>
> **Q1. Clarification on the qualitative results in Figure 4.**
>
> We thank the reviewer for raising clarification on our important qualitative results.
> In Figure 4, we presented two scenarios in two rows. The three columns show the non-safety-critical scenes (left), VAD rollout results under safety-critical scenes (middle), and MPA-VAD rollout results under safety-critical scenes (right). We only pick 2 key frames (pre-crash, and crash frame) for each scenario due to the space limit, highlighting the different outcomes of VAD (crashing) and MPA-VAD (safe) at the end of the evaluation episode. We further illustrate more comprehensive 4-frame qualitative results in Figure 11 and 12 in our supplementary materials. Please kindly check them if interested.
>
> **Q2. According to the original HUGSIM paper, UniAD generally performs better than VAD on HDScore, which is contradictory to the authors results. Could you explain why it is the case?**
>
>
> We believe the difference stems primarily from the evaluation set.
>
> * **Scope of scenes**. The HUGSIM paper reports results on a small, high-quality subset of nuScenes (one scene in their initial release and eight scenes in their latest). To obtain broader coverage, we trained 3DGS reconstructions for more than 360 nuScenes scenes and evaluated on **290 training and 80 for testing** (including safety-critical) cases.
> * **Rendering fidelity and perception sensitivity.** While this larger set offers greater diversity, some of the newly reconstructed scenes could lead to lower perception quality than the hand-selected ones in the HUGSIM release. Compared to UniAD, VAD consumes a vectorized BEV scene representation and is more robust, thus ranking higher on our expanded evaluation set.
>
> Despite these differences, MPA boosts both UniAD and VAD on our extensive benchmark and the original HUGSIM scenes. We will clarify this point in the revision and explore a better balance between scene diversity and rendering fidelity in future work.
>
>
> > [1] Chen, Jianyu, Bodi Yuan, and Masayoshi Tomizuka. "Model-free deep reinforcement learning for urban autonomous driving." IEEE ITSC 2019.
> >
> > [2] Ljungbergh, William, et al. "Neuroncap: Photorealistic closed-loop safety testing for autonomous driving." ECCV 2024.

---

> > ### Comment · Reviewer_2ZCm · 2025-08-07
> >
> > I appreciate the authors' detailed response and decide to keep my score as accept.

---

> > > ### Author Response · Authors · 2025-08-08
> > >
> > > We appreciate the dedicated efforts and the insightful reviews from reviewer 2ZCm. The contributive review helps improve our paper's quality during the revision and rebuttal phase. We will update our revised manuscript according to the suggestions in the review.

---

### Official Review · Reviewer_hzLc · 2025-07-03

**Clarity:** 3
**Significance:** 3
**Originality:** 2
**Rating:** 4
**Confidence:** 2

**Summary:**

The authors present a system designed to adapt open-loop end-to-end driving agents for closed-loop deployment. The proposed framework comprises three key components: (1) counterfactual data generation using 3DGS, (2) a diffusion-based residual policy adaptor to address observation mismatches, and (3) a Q-value model to capture outcomes. The authors identify objective mismatch as the primary cause of closed-loop performance degradation and validate their approach through experiments on the nuScenes dataset.

**Questions:**

In addition to the issues mentioned above:
Since the HDScore is influenced by multiple factors such as RC, NC, and TTC, why is the HDScore particularly high for MPA? Which specific factors contribute most significantly to the reported HDScore?

**Ethical Concerns:**

["NO or VERY MINOR ethics concerns only"]

**Final Justification:**

Thanks for your detailed response. I have raised my score on clarity due to the additional response.

**Limitations:**

yes

**Quality:**

3

**Strengths And Weaknesses:**

Quality
The paper presents a reasonable idea, and the system’s performance is supported by the released experimental results. However, there are several major concerns.
(1) The writing lacks clarity, with insufficient explanation of the motivation and component design details.
(2) The claims in the introduction "the performance drop in close-loop ealuation stems from observation mismatch and objective mismatch " "first mismatch is actually minor in the open-loop evaluation" are not substantiated by evidence or results in the current version.

Clarity
The problem definition is clearly stated. Several concerns remain regarding the motivation and component design details.
(1) the explanation in lines 103–111 is unclear, particularly regarding how the prediction horizon is controlled and how experiments with 3DGS are conducted.
(2) We cannot find evidence Figure/Table to support the claim "the fidelity of ... qualicy."
(3) The r(s_t, a_t) and Sim.step is ambiguous, though it can be inferred that HDSim is used.
(4) the icons used in the MPA column of Figure 4 and in Figure 1 are not well explained, leading to confusion.

Significance
The work is of interest to the industry, and the results demonstrate the effectiveness of the proposed method. Validation on the HDSim environment based on the nuScenes dataset is appropriate. However, the results could be more comprehensive, for example by including experiments on an additional dataset and providing more visualization cases.

Originality

The paper proposes a system that integrates previously explored techniques, such as the use of 3DGS for simulating actions and outcomes (as in RAD) and inference-time scaling with customized rewards. The novelty lies in combining these components into a unified framework and applying them to end-to-end driving agents.

---

> ### Author Rebuttal · Authors · 2025-07-31
>
> We thank Reviewer hzLc for recognizing the well definition and industry-relevance of our work, as well as the effectiveness of our unified MPA framework on the nuScenes closed-loop benchmarks. We answer each question in turn as follows.
>
> **W1. Regarding insufficient motivation and details on individual components.**
>
> Our MPA composes of three major components: (1) counterfactual data generation, (2) diffusion adapter, and (3) Q value network for inference-time scaling.
> 1. The counterfactual data generation aims to provide better data coverage by conducting random sampling based on the behavior policy. The 3DGS model for the training scenes will be used to synthesize our counterfactual observations with action and reward annotations, composing the training data for diffusion adapter and Q value networks.
> 2. The diffusion adapter aims to provide diverse residual action proposals to effectively correct the output actions from pretrained base policy, such as UniAD, VAD, etc.
> 3. The Q-network aims to evaluate the quality of the corrected trajectory proposals from diffusion adapter by different aspects: driveable area compliance, collision avoidance, route following, and speeding constraints.
>
> **W2 & W4. Some claims are not substantiated by evidence or results in the current version.**
>
> We clarify the links between our claims and the accompanying results below:
>
> 1. **"The performance drop in closed-loop evaluation stems from *observation mismatch* and *objective mismatch*."**
>
>    *Evidence:* This statement follows directly from the decomposition in **Eq. (1)**, which analytically separates the two sources of error. The objective-mismatch term captures the gap between the training loss and the online reward, while the observation-mismatch term accounts for perceptual discrepancies introduced by the simulator.
>
> 2. **"…the first mismatch is actually minor in the open-loop evaluation."**
>
>    *Evidence:* Figure 2 supports this point: prediction errors for policies using **ground-truth** visual inputs and those using **3DGS-rendered** inputs are nearly identical, indicating that observation mismatch has negligible impact under open-loop evaluation.
>
> 3. **"the fidelity of the 3DGS-based simulation in its reconstruction quality."**
>
>     *Evidence:* Figure 2 validates that the reconstructed data has minor impact on the prediction error of end-to-end autonomous driving policies, such as UniAD, VAD, and LTF. We further evaluate the rendering quality of novel view synthesis. We can see that the FID scores of 3DGS rendered images are around 20 when the view pose has small lateral displacements with the ground truth trajectories.
>
> We will revise the manuscript to make the presentation more coherent between the analytical derivation and the experimental validation.
>
> **W3. In the preliminary experiment, how is the prediction horizon controlled and how are experiments with 3DGS conducted?**
>
> We appreciate the reviewer’s request for clarification.
>
> **Prediction horizon.** Each base policy produces a sequence of future way-points spanning 3 s. To create the horizons shown in Fig. 2, we truncate that sequence after 1, 2, … H steps and compute the L2 error between the retained way-points and the ground-truth trajectory. Thus the horizon is controlled entirely by how many of the already-predicted way-points we evaluate—no additional forward passes are required.
>
> **3DGS experiments.** Following the HUGSIM protocol, we train a separate 3D Gaussian-Splatting (3DGS) model for every scene. Ground, static background, and dynamic objects are reconstructed as three disjoint Gaussian sets and composited at render time. In the preliminary evaluation, we run each policy twice: (1) on the original nuScenes camera images and (2) on the images rendered by the corresponding 3DGS model. Then we measure prediction errors with the same horizon-truncation procedure described above.
>
> **W5. What is the meaning of $r(s_t, a_t)$ and `env.step()` in the pseudo code?**
>
> We apologize for the insufficient explanation in the current manuscript. Based on our MDP's definition in section 2.1, $r(s_t, a_t)$ is the reward feedback that HUGSIM returns to evaluate the performance based on the current driving states and actions. `env.step()` is called to update the next vehicle states as well as the corresponding visual observations with HUGSIM's 3DGS rendering engine. The collected counterfactual dataset is composed of (s, a, o, r), including the state action transition pairs, visual observations and the reward labels. They will later be used to train the policy adapter and value model.
>
> **W6. Icon explanation in Figure 1 and 3.**
>
> We thank the reviewer for pointing out this concern. The icons in Figure 1 and 3 include the driving actions (throttle and steering), synthetic counterfactual dataset, frozen base E2E policies, as well as the trainable diffusion adapter and value networks. We will add a legend in our revised manuscript to help the audience better understand our pipeline details.
>
>
> **W7. Additional visualization and more comprehensive results on different datasets.**
>
> **Additional visualization.** Beyond Figure 4, we offer additional visualizations for all nine evaluated methods across four distinct scenarios in Appendix Figures 11 and 12. Each scenario is illustrated with four representative key frames, covering both safety-critical and non-safety-critical traffic contexts. We will include more qualitative examples in the appendix for our final revised version.
>
> **Additional baseline comparison.** We add additional baseline results on imitation learning with data aggregation (BC-Safe-DAgger), and model-based RL agent (LAW-Imagine).
> * DAgger [1] is an online imitation learning approach that extends our BC-Safe baseline in aggregating new online rollout data. We observe that BC-Safe-DAgger performs better compared to BC-Safe, particularly in the safety-critical scenes. But it still cannot outperform our proposed MPA (reported in Table 1 in the original manuscript).
> * LAW [2] follows VAD and incorporates additional latent world model loss as an auxiliary regularization when predicting actions during pretraining. We use the output trajectory from LAW as the base action, and further incorporate imagination rollout in MBRL using the same value functions as MPA for LAW, denoted as **LAW-Imagine**. We use LAW's checkpoint from their public codebase, and report the results of LAW-Imagine on HUGSIM in Table R1. The results show that with world model augmentation and added imagination rollout, **LAW-Imagine** can outperform three pretrained base policies -- UniAD, VAD, and LTF, but is still not as good as MPA (reported in Table 1 in the original manuscript) due to the significant drop in the comfort of the driving behaviors.
>
> **Table R1.1: Additional in-domain results**
> |Method|RC|NC|DAC|TTC|COM|HDScore|
> |-|-|-|-|-|-|-|
> |BC-Safe|57.0|59.8|87.9|55.2|89.4|33.6|
> |**BC-Safe-DAgger**|63.4|64.6|83.3|53.9|94.9|33.0|
> |**LAW-Imagine**|77.3|84.7|94.6|82.6|41.8|50.9|
>
> **Table R1.2: Additional results under unseen scenes**
> |Method|RC|NC|DAC|TTC|COM|HDScore|
> |-|-|-|-|-|-|-|
> |BC-Safe|59.2|59.8|81.2|56.3|95.9|34.6|
> |**BC-Safe-DAgger**|57.6|61.9|79.0|54.1|94.5|30.5|
> |**LAW-Imagine**|79.5|79.6|96.6|78.3|38.6|49.5|
>
> **Table R1.3: Additional results under safety-critical scenes**
> |Method|RC|NC|DAC|TTC|COM|HDScore|
> |-|-|-|-|-|-|-|
> |BC-Safe|20.2|80.1|91.7|67.3|86.7|13.5|
> |**BC-Safe-DAgger**|35.9|78.4|93.3|66.8|99.7|17.6|
> |**LAW-Imagine**|71.7|85.1|98.4|81.7|44.8|50.5|
>
> **Additional dataset evaluation.** Besides additional baselines, we also evaluate MPA’s effectiveness on the Waymo dataset as well as on nuScenes. The Waymo dataset contains only three front-facing camera views, whereas nuScenes provides six. Consequently, we report the zero-shot performance of the nuScenes-trained LTF and MPA (LTF) models, which accept three-camera inputs. We evaluate on the first 40 scenes of the “NeRF on the Road” (NOTR) dataset used in EmerNeRF [3]. The results below demonstrate that MPA generalizes well and delivers consistent performance gains on datasets beyond nuScenes.
>
>
> |Method|RC|NC|DAC|TTC|COM|HDScore|
> |-|-|-|-|-|-|-|
> |LTF|0.48|0.67|0.54|0.62|0.61|0.24|
> |MPA (LTF)|0.71|0.76|0.65|0.71|0.71|0.37|
>
>
>
> **Q1. Which specific factors contribute most significantly to the reported HDScore?**
>
> As we stated in the paper, the HDScore is computed as:
>
> $$
> \\text{HDScore}=\\text{RC} \\times \\frac{1}{T}\\sum_{t=0}^T \\Big\\{ \\prod_{m\\in \\{\\text{NC, DAC}\\}} \\text{score}_m \\times  \frac{\sum\_{m \\in \\{\\text{TTC, COM}\\}}  \\text{weight}_m   \\times \\text{score}_m}{\sum\_{m\\in \\{\\text{TTC,COM} \\}} \\text{weight}_m} \\Big\\}_t.
> $$
>
> Our MPA method prevails baselines particularly in the **Route Completion (RC)** and **Driveable Area Compliance (DAC)** metrics, which leads to a higher HDScore. Please kindly check Tables 1 and 2 in our original manuscript for more details.
>
> > [1] Ross, Stéphane, et al. "A reduction of imitation learning and structured prediction to no-regret online learning." AISTATS 2011
> >
> > [2] Li, Yingyan, et al. "Enhancing End-to-End Autonomous Driving with Latent World Model." ICLR 2025.
> >
> > [3] Yang, Jiawei, et al. "EmerNeRF: Emergent Spatial-Temporal Scene Decomposition via Self-Supervision." ICLR 2024

---

### Official Review · Reviewer_s9Gr · 2025-07-06

**Clarity:** 3
**Significance:** 2
**Originality:** 2
**Rating:** 3
**Confidence:** 5

**Summary:**

The paper presents Model4 based Policy Adaptation (MPA), a general framework that attempts to fortify pretrained end-to-end driving policies against distributional shift. MPA first uses a simulator to create many counterfactual/ “what-if” driving situations the original data never saw. Using this synthetic corpus, it jointly learns a diffusion-based policy adapter that refines the agent’s raw trajectory proposals and a multi-step Q-value estimator that forecasts the long-horizon utility of each candidate. At inference, the adapter produces multiple feasible trajectories and the Q-value network selects the one with the greatest expected return, thereby delivering safer, more robust driving behavior without retraining the underlying policy.

**Questions:**

Please see weaknesses section

**Ethical Concerns:**

["NO or VERY MINOR ethics concerns only"]

**Final Justification:**

Based on the discussion with the authors and the constructive exchange with AC, I have decided to maintain my initial rating of 3, as certain components remain insufficiently clear. However, I completely understand and have no objection if the paper is accepted.

**Limitations:**

Yes

**Quality:**

2

**Strengths And Weaknesses:**

**Strengths:**

1. The paper tackles model drift in closed-loop driving, where small deviations compound over time and cause severe distribution shift. The topic is both important and under active investigation.

2. The writing is generally clear, the pipeline is easy to follow, and figures are helpful.

**Weaknesses and questions:**

1. Combining synthetic counterfactual data with a learned value head is well-established in model-based RL. The contribution appears to be an incremental recombination of known pieces rather than a fundamentally new idea.

2. The motivation for all the submodules (simulator, diffusion adapter, Q-network) is not clear.


3. Why not have the Q-value network directly on the counterfactual trajectories and select among them, bypassing the diffusion adapter? What advantage does the diffusion adapter offer?


4. I missed mention and comparison to key model-based RL literature (e.g., PlaNet, Dreamer, MBRL for autonomous driving).


5. Algorithms addressing compounding error by online data aggregation (e.g., DAgger) or by combining imitation with value optimization are absent.


6. I missed ablation to some of the components (aligned with comment 2 and 3).


7. There are quite some similar recent works aligning with the model-based counterfactual data generation for the same problem setting. These are not cited.

**References:**

1.	Hafner, Danijar, et al. "Dream to control: Learning behaviors by latent imagination." arXiv preprint arXiv:1912.01603 (2019).

2.	Hafner, Danijar, et al. "Mastering atari with discrete world models." arXiv preprint arXiv:2010.02193 (2020).

3.	Ross, Stéphane, Geoffrey Gordon, and Drew Bagnell. "A reduction of imitation learning and structured prediction to no-regret online learning." Proceedings of the fourteenth international conference on artificial intelligence and statistics. JMLR Workshop and Conference Proceedings, 2011.

4.	Dey, S., Paassen, B., Nair, S. R., Boughorbel, S., & Schilling, A. F. (2024). Continual Learning from Simulated Interactions via Multitask Prospective Rehearsal for Bionic Limb Behavior Modeling. arXiv preprint arXiv:2405.01114.

---

> ### Author Rebuttal · Authors · 2025-07-31
>
> We express our gratitude to reviewer s9GR for their feedback and for acknowledging our contribution in closed-loop autonomous driving and clear presentation in the methodology pipeline. We provide our response to the questions as below.
>
> **Q1. Regarding the 'incremental recombination of known pieces'.**
>
> We thank the reviewer for raising this clarification point regarding our major contribution. While pairing synthetic data with a value head is well-explored in MBRL, our proposed MPA addresses an open challenge in the real world problem: photorealistic end-to-end closed-loop driving. MPA boosts HDScore over UniAD/VAD/LTF in a scalable evaluation pipeline using nuScenes dataset. To our knowledge, no prior MBRL or E2E driving work unifies sensor-level counterfactual generation, diffusion-based residual control, and online multi-principle value guidance in a single real-time system. We believe that MPA could offer a meaningful advance in E2E driving problem beyond an incremental recombination.
>
> **Q2 & Q6. The motivation for all the submodules (simulator, diffusion adapter, Q-network) is not clear. The corresponding ablation results (on these submodules) are unclear.**
>
> **Motivation of each module:** We thank the authors for raising this important aspect of our key methodology.
>
> * The adopted simulator HUGSIM provides a systematic benchmark for closed-loop end-to-end autonomous driving, which is missing in the preexisting open-loop evaluation in end-to-end autonomous driving research. The 3DGS model for the training scenes will be used to synthesize our counterfactual observations with action and reward annotations, composing the training dataset of MPA.
> * The diffusion adapter aims to provide diverse residual action proposals to effectively correct the output actions from pretrained base policy, such as UniAD, VAD, etc.
> * The Q-network aims to evaluate the quality of the corrected trajectory proposals from diffusion adapter by different aspects: driveable area compliance, collision avoidance, route following, and speeding constraints.
>
> **Ablation results of each module:** The effectiveness of counterfactual data generation is visualized in Figure 5, where more counterfactual rollout steps yield to better final performance. The ablation studies in Table 3 of our original manuscript show the effectiveness of Q value function guidance and the policy adapter.
>
> **Q3. What advantage does the diffusion adapter offer?**
>
> As introduced in Q2, the diffusion adapter offers more diverse and high-quality proposals of action correction, which are learned based on the behavior from counterfactual data.  This effect of the policy adapter is demonstrated in the ablation studies of Table 3 in our original manuscript, where method without policy adapter (ID-5) suffered performance degradation compared to the full model, especially under safety-critical scenarios.
>
> **Q4. Discussion and comparison with Model-Based RL approaches.**
>
> We thank the reviewer for raising the line of model-based RL research. World model indeed plays an important role in end-to-end autonomous driving. We will add more discussion and references in our revised manuscripts with existing MBRL and works like PlaNet, Dreamer as the reviewer mentioned.
>
> Besides the current baselines, we compare our method with a latent world model-based end-to-end driving framework LAW [1]. LAW follows VAD and incorporates additional latent world model loss as an auxiliary regularization when predicting actions during pretraining. We use the output trajectory from LAW as the base action, and further incorporate imagination rollout in MBRL using the same value functions as MPA for LAW, denoted as **LAW-Imagine**. We use LAW's checkpoint from their public codebase, and report the results of LAW-Imagine on HUGSIM in **Table R1.1-R1.3**.
> The results show that with world model augmentation and added imagination rollout, **LAW-Imagine** can outperform three pretrained base policies UniAD, VAD, and LTF in route completion, and it prevails in non-collision and time-to-collision. However, the comfort of LAW-Imagine is the far worse than the other approaches, which means the agents possibly driving in a zig-zag manner to guarantee safety. As a result, the Route completion and HDScore of LAW-Imagine is not as good as MPA-enhanced base policies.
>
>
> **Q5. Comparison with imitation learning with online data aggretation.**
>
> We extend our BC-Safe baseline [2] with **online data aggregation** [3], i.e. rolling out new counterfactual dataset based on the trained policy and aggregate them to the next round of policy learning. We conduct one round of such data aggregation on the training dataset, and report the performance of **BC-Safe-DAgger** for both training (in-domain), testing (unseen scenes), and safety-critical scenes in **Table R1.1-R1.3**. The results show that data aggregation helps improve BC-Safe's route completion performance for in-domain setting and safety-critical setting, but has limited improvements in the unseen scenes.
>
> **Table R1.1: Additional in-domain results and comparison with existing base policies with MPA. Bold means the best for individual metrics.**
> |Method|RC|NC|DAC|TTC|COM|HDScore|
> |-|-|-|-|-|-|-|
> |BC-Safe|57.0|59.8|87.9|55.2|89.4|33.6|
> |**BC-Safe-DAgger (New)**|63.4|64.6|83.3|53.9|94.9|33.0|
> |**LAW-Imagine (New)**|77.3|**84.7**|94.6|**82.6**|41.8|50.9|
> |UniAD|39.4|56.9|75.1|52.1|**98.7**|19.4|
> |VAD|50.1|68.4|87.2|66.1|90.2|31.9|
> |LTF|65.2|71.3|92.1|67.6|98.4|46.7|
> |MPA (UniAD)|93.6|76.4|92.8|72.8|91.8|66.4|
> |MPA (VAD)|**94.9**|75.4|**93.6**|72.5|92.8|**67.0**|
> |MPA (LTF)|93.1|70.8|90.9|67.9|94.9|60.0|
>
> **Table R1.2: Additional results under unseen scenes and comparison with existing base policies with MPA. Bold means the best for individual metrics.**
> |Method|RC|NC|DAC|TTC|COM|HDScore|
> |-|-|-|-|-|-|-|
> |BC-Safe|59.2|59.8|81.2|56.3|95.9|34.6|
> |**BC-Safe-DAgger (New)**|57.6|61.9|79.0|54.1|94.5|30.5|
> |**LAW-Imagine (New)**|79.5|**79.6**|96.6|**78.3**|38.6|49.5|
> |UniAD|39.3|56.6|74.0|52.6|**98.2**|22.2|
> |VAD|45.4|64.8|86.2|62.0|95.9|29.3|
> |LTF|63.3|64.8|86.5|62.8|**98.2**|41.9|
> |MPA (UniAD)|**93.7**|69.5|92.9|66.6|97.6|60.9|
> |MPA (VAD)|90.9|71.0|**94.4**|68.8|97.7|**61.2**|
> |MPA (LTF)|91.8|68.3|91.0|66.5|96.9|57.0|
>
> **Table R1.3: Additional results under safety-critical scenes and comparison with existing base policies with MPA. Bold means the best for individual metrics.**
> |Method|RC|NC|DAC|TTC|COM|HDScore|
> |-|-|-|-|-|-|-|
> |BC-Safe|20.2|80.1|91.7|67.3|86.7|13.5|
> |**BC-Safe-DAgger (New)**|35.9|78.4|93.3|66.8|99.7|17.6|
> |**LAW-Imagine (New)**|71.7|**85.1**|98.4|**81.7**|44.8|50.5|
> |UniAD|11.4|76.2|82.1|57.8|95.9|4.5|
> |VAD|25.4|77.0|88.3|73.2|88.4|16.0|
> |LTF|35.1|80.9|96.8|78.1|**100.0**|24.2|
> |MPA (UniAD)|95.1|76.8|98.9|74.2|97.7|70.4|
> |MPA (VAD)|**96.6**|79.8|**99.0**|77.3|97.7|**74.7**|
> |MPA (LTF)|87.3|72.0|94.0|66.9|97.8|56.3|
>
> **Q7. Missing literature review on counterfactual data augmentation in Model-Based RL.**
>
> We thank the authors for raising these related works, indeed some of the counterfactual data augmentation works contribute to the generalizability of MBRL agents, e.g. [4, 5]. We will add the discussion on them in the related work in our revised manuscript.
>
>
> > [1] Li, Yingyan, et al. "Enhancing End-to-End Autonomous Driving with Latent World Model." ICLR 2025.
> >
> > [2] Pan, Yunpeng, et al. "Agile autonomous driving using end-to-end deep imitation learning." RSS 2018.
> >
> > [3] Ross, Stéphane, et al. "A reduction of imitation learning and structured prediction to no-regret online learning." AISTATS 2011.
> >
> > [4] Pitis, Silviu, Elliot Creager, and Animesh Garg. "Counterfactual data augmentation using locally factored dynamics." NeurIPS 2020.
> >
> > [5] Pitis, Silviu, et al. "Mocoda: Model-based counterfactual data augmentation." NeurIPS 2022.

---

> > ### Comment · Reviewer_s9Gr · 2025-08-06
> >
> > I thank the authors for their rebuttal. However, I remain unconvinced by the response, and it does not substantially alter my assessment of the paper. My main concerns are as follows:
> >
> > **1. Method Novelty:** My initial understanding is reinforced by the authors’ own response: the proposed method appears to be a straight combination of existing components (MBRL+ diffusion adapter) rather than a fundamentally novel contribution.
> >
> >
> > **2. Need for Diffusion Adapter:** If I interpret Table 3 correctly, in 4 out of 6 unseen scene scenarios, the model **without** the diffusion adapter performs better than or similarly to the version with it. In safety-critical scenarios, a similar trend is observed in 3 out of 6 cases. In one example, the reported performance difference is only 0.1 (98.9 vs. 98.8), which could easily be attributed to variance from random seeds. Thus, the necessity and benefit of the diffusion adapter are not convincingly demonstrated.
> >
> >
> > **Baselines:** It remains unclear which baselines are newly introduced and which were already included in the original submission. From the rebuttal, it appears that only one new baseline (LAW) has been added. However, key prior works such as Dreamer, PlaNet, and other model-based RL approaches are still missing from the comparisons. The rebuttal includes additional tables (e.g., R1.2, R1.3) focusing on variants of the authors’ own policies. It is unclear which reviewer comment these address. Instead of comparing against strong, established baselines, the rebuttal adds more variations of the authors’ own models, which seems to inflate the results without providing clearer insights.

---

> > > ### Author Response · Authors · 2025-08-07
> > >
> > > We thank the reviewer for the follow-up response. We'd like to address their remaining concerns concisely.
> > >
> > > **Method novelty.** MPA is the first plug-and-play adapter that jointly (i) generates photorealistic counterfactual rollout data with a fidelity-versus-diversity tradeoff, (ii) applies a residual-vector diffusion adapter to refine base trajectories with diverse proposals, and (iii) guides inference with a multi-step, multi-objective value network trained on the same imagination rollouts. This integration demanded stable diffusion adapter training and joint optimization of four Q-heads so that safety, comfort, and route-following improve simultaneously. Such effort is absent from prior E2E-driving literature.
> > >
> > >
> > > **Ablation of the diffusion adapter.**  In Table 3, the six columns represent our evaluation metrics (RC, NC, DAC, TTC, COM, HDScore), but **not individual scenes**. The precise definition of these scores follow prior works [1, 2]. We report the average score over 70 unseen and 10 safety-critical nuScenes scenes. Removing the diffusion adapter reduces route completion by 2.6 % on unseen and 19.5 % on safety-critical cases, and drops HDScore by 15.1 points (out of 100). Figure 9 in the appendix further shows that increasing the adapter’s mode count consistently boosts RC, DAC and HDScore, underscoring the value of diverse residual corrections.
> > >
> > > **Baseline selection.** LAW is the only open-source world-model-based E2E driving codebase that takes multi-camera inputs and outputs continuous trajectories, so we adopt it as our model-based baseline. Prior Visual MBRL works, such as Dreamer and PlaNet, target on Atari/MuJoCo tasks and have no prior E2E driving implementation, so evaluating them directly in E2E driving task is infeasible given the limited time of the rebuttal. Still, we integrated LAW into the HUGSIM simulator and added imagination rollouts ("LAW-Imagine") for a fair comparison. Tables R1.1–R1.3 show that while LAW-Imagine improves NC and TTC, it still trails MPA on the composite HDScore because of lower comfort, confirming that MPA’s gains are not due to the weak baselines.
> > >
> > > We hope this clarifies our contribution, the diffusion-adapter evidence, and the rationale for our baseline selection.
> > >
> > > > [1] Dauner, Daniel, et al. "Navsim: Data-driven non-reactive autonomous vehicle simulation and benchmarking." NeurIPS D&B Track 2024
> > > >
> > > > [2] Zhou, Hongyu, et al. "Hugsim: A real-time, photo-realistic and closed-loop simulator for autonomous driving." arXiv preprint arXiv:2412.01718 (2024).

---

> ### Author Response · Authors · 2025-08-09
>
> Dear Reviewer s9Gr,
>
> Thank you for engaging in our discussion. Since the discussion period is coming to an end, we'd like to see if you have any additional feedback regarding our previous response. Thank you again for your time during the review and discussion period.
>
> Best,
>
> Authors of Submission #26821

---

### Decision · Program_Chairs · 2025-09-17

**Decision:**

Accept (poster)

**Comment:**

This paper introduces Model-based Policy Adaptation (MPA), a framework to improve the robustness of end-to-end autonomous driving policies under distributional shift. A key challenge arises from objective mismatch, as offline imitation learning lacks reward feedback, leading to degraded closed-loop performance. To address this, MPA leverages counterfactual data generation using 3D Gaussian Splatting (3DGS) to create novel driving scenarios unseen during training. A diffusion-based policy adapter refines trajectory proposals by predicting residual corrections for the pretrained policy. In parallel, a multi-step Q-value model evaluates long-horizon outcomes to select the safest and most effective trajectory at inference. The combined system adapts open-loop policies for closed-loop deployment without retraining the original backbone. Experiments demonstrate substantial performance gains, particularly in safety-critical situations.

The rewievers were all very interested in the problem and the proposed approach. They asked many questions, in particular about the validation of the method based on the various metrics proposed and the types of scenarios.
At the end of the rebuttal, where the authors provided many detailed answers and clarifications on their approach, further discussions with the AC cleared up some misunderstandings, and even if some reservations remain on the clarity of the gains that do not emerge for all the proposed metrics, the AC proposes to accept this submission, following the very positive opinions of 3 of the 4 Rs.